# E²RANK: YOUR TEXT EMBEDDING CAN ALSO BE AN EFFECTIVE AND EFFICIENT LISTWISE RERANKER

## ABSTRACT

Text embedding models serve as a fundamental component in real-world search applications. By mapping queries and documents into a shared embedding space, they deliver competitive retrieval performance with high efficiency. However, their ranking fidelity remains limited compared to dedicated rerankers, especially recent LLM-based listwise rerankers, which capture fine-grained query-document and document-document interactions. In this paper, we propose a simple yet effective unified framework **E²RANK**, means **E**fficient **E**mbedding-based **Rank**ing (also means **Embedding-to-Rank**), which extends a single text embedding model to perform both high-quality retrieval and listwise reranking through continued training under a listwise ranking objective, thereby achieving strong effectiveness with remarkable efficiency. By applying cosine similarity between the query and document embeddings as a unified ranking function, the listwise ranking prompt, which is constructed from the original query and its candidate documents, serves as an enhanced query enriched with signals from the top-K documents, akin to pseudo-relevance feedback (PRF) in traditional retrieval models. This design preserves the efficiency and representational quality of the base embedding model while significantly improving its reranking performance. Empirically, E²RANK achieves state-of-the-art results on the BEIR reranking benchmark and demonstrates competitive performance on the reasoning-intensive BRIGHT benchmark, with very low reranking latency. We also show that the ranking training process improves embedding performance on the MTEB benchmark. Our findings indicate that a single embedding model can effectively unify retrieval and reranking, offering both computational efficiency and competitive ranking accuracy.

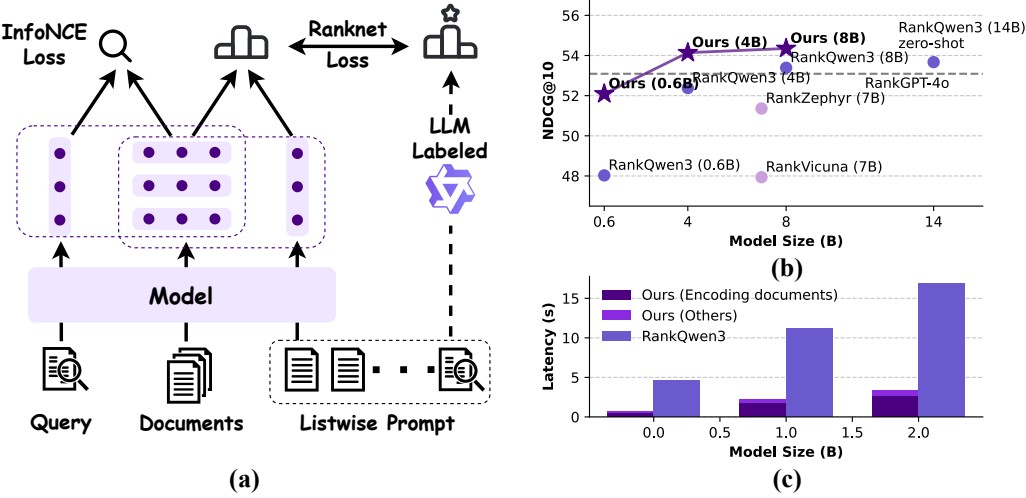

Figure 1: **(a)** Overview of E²RANK. **(b)** Average reranking performance on the BEIR benchmark, E²RANK outperforms other baselines. **(c)** Reranking latency per query on the Covid dataset.

# 1 INTRODUCTION

Text embedding and reranking are fundamental components in numerous natural language processing (NLP) and information retrieval (IR) applications, including web search, question answering, retrieval-augmented generation, and beyond (Karpukhin et al., 2020; Zhao et al., 2024). In general, most production IR systems adopt a two-stage architecture: a lightweight embedding retriever retrieves a small candidate set, which is then reranked by a more powerful reranking model (Matveeva et al., 2006). In the first stage, text embedding offers efficient similarity search abilities by mapping queries and documents into a shared low-dimensional vector space, enabling real-time and web-scale applications (Karpukhin et al., 2020). The advent of large language models (LLMs) has further improved the retrieval performance of these embedding models (Zhu et al., 2023; Ma et al., 2023; BehnamGhader et al., 2024).

However, a performance gap persists between embedding-based retrievers and state-of-the-art rerankers, particularly those using LLMs (Zhu et al., 2023). Specifically, listwise methods like RankGPT (Sun et al., 2023) can model fine-grained interactions within the entire candidate set and capture both query-document and document-document relationships, leading to rankings that better reflect human judgment and achieving state-of-the-art results across various benchmarks (Sun et al., 2023; Pradeep et al., 2023).

Despite their effectiveness, LLM-based listwise rerankers incur high computational costs and inference latency, limiting their deployment in real-time environments. The need to encode all candidates in a single pass introduces substantial prefilling delays, while autoregressive decoding further slows the process (Liu et al., 2025b). Therefore, some recent works tried to improve the efficiency of listwise rerankers by compressing the input documents (Liu et al., 2025b) and leveraging LLM's output logits or attention patterns to avoid expensive auto-regressive generation (Reddy et al., 2024; Chen et al., 2024b; Zhang et al., 2025b).

Among the above works, an important observation is that the auto-regressive generation paradigm adopted by RankGPT (Sun et al., 2023) is not necessary for ranking, while the interaction between query and documents in the context is critical for ranking effectiveness (Chen et al., 2024b). Additionally, Liu et al. (2025b) shows that incorporating document embeddings in the ranking process is also helpful. Based on these, we naturally raise the following question: *What if incorporating the interaction signals in embedding models for reranking?*

Intuitively, this question can be addressed from two complementary perspectives. From the standpoint of dense retrieval, the listwise prompt integrating both the document and the query can be viewed as a form of pseudo relevance feedback (PRF) (Xu & Croft, 1996) query in traditional IR, which can enhance the quality of query embeddings (Yu et al., 2021). Conversely, from the perspective of listwise reranking, the rich contextual information encoded in the listwise prompt enables the use of simple cosine similarity in place of autoregressive decoding. In essence, the listwise prompt can be transformed into a single PRF-enhanced query embedding, allowing reranking to be efficiently performed via cosine similarity against precomputed document embeddings. This leads to a unified scoring mechanism and a unified model that seamlessly bridges retrieval and reranking.

We then introduce $E^2$RANK (**E**fficient **E**mbedding-based **Rank**ing or **Embedding-to-Rank**) and propose a two-stage process to train the unified model, shown in Figure 1. First, we train an embedding model via contrastive learning, then continue to train it under a multi-task learning framework that jointly optimizes contrastive and ranking objectives. Specifically, we use the listwise prompt as a pseudo query and adopt the RankNet loss (Burges et al., 2005) for optimization. This multi-task approach encourages the embedding space to capture both query-document relevance and full interactions. At inference time of reranking, we only compute cosine similarity between document embeddings and the optimized query representation derived from the listwise prompt. This unified design offers advantages for both efficiency and effectiveness. First, by operating in the embedding space instead of generation, it eliminates the computational overhead of LLM-based rerankers, enabling low-latency inference suitable for large-scale applications. Second, the full interaction between query and documents and richer training signals substantially enhances reranking quality.

We evaluate $E^2$RANK on popular reranking and embedding benchmarks. Experimental results demonstrate that our model achieves state-of-the-art reranking performance on BEIR (Thakur et al., 2021) and exhibits a strong performance on reasoning-intensive benchmark BRIGHT (Su et al.,

2025), while notably improving inference efficiency. Additionally, trained solely on public data, our model preserves a competitive embedding performance on MTEB (Muennighoff et al., 2022), demonstrating the effectiveness of unifying retrieval and reranking.

Our contributions are summarized as follows:

- We reinterpret the listwise prompt as a PRF query and propose a unified framework, $E^2$RANK, for both retrieval and reranking.
- We propose a two-stage training process to optimize the unified model for both retrieval and listwise reranking tasks.
- Extensive experiments show that $E^2$RANK achieves state-of-the-art reranking performance, with significantly lower latency than existing LLM-based rerankers, while maintaining competitive retrieval performance on MTEB.

## 2 RELATED WORK

**Large Language Model For Document Reranking**  Large language models (LLMs) like GPT-4 (OpenAI, 2024) and Qwen-3 (Yang et al., 2025) have significantly advanced information retrieval, achieving state-of-the-art performance in document ranking tasks across multiple benchmarks (Sun et al., 2023; Zhu et al., 2023; Chen et al., 2024c). Existing methods generally fall into three prompting paradigms: pointwise, pairwise, and listwise. Pointwise methods evaluate each query-document pair independently, offering efficiency but lacking cross-document comparisons (Liang et al., 2022; Sachan et al., 2022; Zhang et al., 2023a; Liu et al., 2024b). Pairwise methods compare document pairs for a given query to determine relative relevance (Qin et al., 2023). Listwise methods instead consider the entire candidate set simultaneously and generate a ranking list based on global relevance signals (Sun et al., 2023; Pradeep et al., 2023; Liu et al., 2024a). Recent studies further improve listwise reranking by refining prompting strategies or the method of outputting the ranking list (Reddy et al., 2024; Liu et al., 2025b; Chen et al., 2024b; Zhang et al., 2025b).

**Text Embedding Models**  Text embeddings map queries and documents into a shared semantic space and serve as a foundation component in modern search systems. Based on pre-trained language models such as BERT (Devlin et al., 2018) and T5 (Raffel et al., 2020), they significantly improved retrieval performance over traditional methods (Karpukhin et al., 2020; Ni et al., 2021; Zhao et al., 2024), and approaches like GTE (Li et al., 2023b), E5 (Wang et al., 2022), and BGE (Xiao et al., 2023) further boosted quality via large-scale contrastive learning. More recently, LLMs have emerged as powerful backbones due to their strong semantic understanding and generalization capabilities. Representative methods include LLM2Vec (BehnamGhader et al., 2024), E5-Mistral (Wang et al., 2023), NV-Embed (Lee et al., 2025), and Qwen3-Embedding (Zhang et al., 2025c), which explore architectural modifications, training data construction, or advanced training strategies. Instruction following and in-context learning abilities of text embeddings are also studied (Su et al., 2022; Li et al., 2024a). Additionally, GritLM (Muennighoff et al., 2024) unified the embedding model and generative model through multi-task learning.

Compared to previous work, our work unifies the embedding and listwise reranking ability, which share a similar objective, in a single embedding model, considering both effectiveness and efficiency.

**Pseudo Relevance Feedback for Dense Retrieval**  Pseudo Relevance Feedback (PRF) is an important concept in classic IR. Specifically, it is an automatic query expansion technique widely used in classic IR (Xu & Croft, 1996; Manning, 2008). After an initial retrieval, the system assumes that the top-K retrieved documents are relevant, extracts informative terms from these documents, and uses them to expand the original query for a second round of retrieval. Recent studies show the effectiveness of incorporating PRF in dense retrievers. ANCE-PRF (Yu et al., 2021) consumed the query and the top retrieved documents to learn a better query encoder, but is less robust for strong models (Li et al., 2022; 2023a). Other works leveraged PRF in rerankers, but were limited in pointwise cross-encoders and needed to generate keywords for query expansion (Li et al., 2024b; Weller et al., 2024). Compared to previous work, we first interpret and systematically study PRF in the framework of LLM-based listwise reranking instead of merely retrieval and without additional query-augmented techniques, and also demonstrate its effectiveness in this context through training under a ranking objective.

## 3 METHODOLOGY

We first review embedding-based retrieval and LLM-based listwise reranking, then present our key insight: listwise prompts can be treated as *pseudo relevance feedback queries*, then the cosine similarity of embeddings could be a unified ranking function, leading to a unified model $\text{E}^2\text{RANK}$. Finally, we detail the training of $\text{E}^2\text{RANK}$.

### 3.1 PRELIMINARY

For a LLM-based decoder-only text embedding model $f$ and any document $d$, we append the special end-of-sequence token [EOS] at the end of the input sequence, and the hidden state at the position of [EOS] from the final decoder layer is taken as the sequence embedding: $\boldsymbol{e}^d = f(d, [\text{EOS}])[-1]$. Further, given a query $q$, we append the instruction $I$ in front of the query to ensure its instruction-following abilities (Su et al., 2022) and obtain the embedding $\boldsymbol{e}^q = f(I, q, [\text{EOS}])[-1]$. The relevance between the query and the document is measured by the cosine similarity between their corresponding embeddings, denoted as $s(q, d) = \cos(\boldsymbol{e}^q, \boldsymbol{e}^d)$.

While the embedding model learning encodes the semantic information of a *single* document in the embedding space, it has not been optimized to capture nuanced differences between *multiple* documents. In contrast, LLM-based listwise rerankers (e.g., RankGPT (Sun et al., 2023)) use a *listwise prompt* that includes the query and the entire candidate set, formulated as $\hat{q} = (I, d_1, ..., d_k, q)$, where $\{d_i\}_{i=1}^k$ is candidate documents set. The model is then asked to output a text form permutation (e.g., "[2] > [1] > [3]...") of the documents in decreasing order of relevance. While effective, this approach requires auto-regressive decoding or full-sequence encoding over long inputs, leading to high computational cost and latency. Moreover, the decoding process is inherently sequential and difficult to parallelize. Meanwhile, some work proposed that the auto-regressive decoding may not be necessary for listwise reranker; however, the listwise prompt containing the interaction between query and documents in the context is the most important (Chen et al., 2024b; Zhang et al., 2025b).

### 3.2 LISTWISE PROMPTS AS PSEUDO RELEVANCE FEEDBACK QUERY

Inspired by these observations, we propose to reinterpret the listwise prompt as a *pseudo-relevance feedback (PRF)* query. Therefore, we can formulate the listwise reranking and retrieval in a *unified* framework. Specifically, instead of generating a ranking list auto-regressively, we start from an embedding model and use the cosine similarity of embeddings as a unified ranking function for both retrieval and reranking. Formally, for the listwise prompt, we obtain its embedding

$$\boldsymbol{e}^{\hat{q}} = f(I, d_1, ..., d_k, q)[-1], \tag{1}$$

and compute $s(\hat{q}, d_i) = \cos(\boldsymbol{e}^{\hat{q}}, \boldsymbol{e}^{d_i})$ as the score for reranking. The instructions we use is similar to "*Given a query and some relevant documents, rerank the documents*", detailed in the Appendix C. It should be noted that, different from text embedding, we apply chat templates for listwise prompt.

This design allows us to exploit listwise information for effectiveness without sacrificing efficiency at inference time. First, the listwise prompt provides the model with additional contextual PRF signals, allowing it to refine the query representation by implicitly leveraging document-document and query-document relationships. Second, both retrieval and reranking reduce to simple cosine similarity computations in the shared embedding space, and the document embeddings can be reused. Finally, PRF-based design enables feeding only partial candidates in LLM inputs for the full ranking, for example, including only top-20 documents in the PRF query to rerank top-100, which can further improve the efficiency.

### 3.3 TRAINING THE UNIFIED EMBEDDING AND LISTWISE RERANKING MODEL

We propose training $\text{E}^2\text{RANK}$ in two stages: first, training an embedding model, then endowing it with listwise reranking capacity.

**Stage I** We start from training an LLM-based decoder-only text embedding model. In the training process, we employ standard contrastive learning to align relevant query–document pairs while pushing apart irrelevant ones. Specifically, for a training query $q_i$, there is one positive document

$d_i^+$ and a set of negative documents $D^-$. Given a batch of $N$ instances, we minimize the InfoNCE loss (Izacard et al., 2021):

$$\mathcal{L}_{\text{InfoNCE}} = -\frac{1}{N} \sum_{i=1}^{N} \log \frac{e^{(s(q_i, d_i^+)/\tau)}}{e^{(s(q_i, d_i^+)/\tau)} + \sum_{d_j \in D^-} e^{(s(q_i, d_j)/\tau)}}, \quad (2)$$

where $\tau$ is a temperature hyperparameter, which is set to 0.03 during training. This embedding training stage ensures that the base embedding model learns strong semantic representations suitable for large-scale retrieval.

**Stage II** To incorporate the listwise reranking capabilities into the embedding model, we continue training the model using a multi-task learning framework. Basically, we include the contrastive learning with InfoNCE loss to maintain the embedding capacity of the model and a new learning-to-rank loss function, RankNet (Burges et al., 2005) loss, which is a pairwise loss that measures the correctness of relative orders, for listwise ranking ability. The RankNet loss is defined as follows:

$$\mathcal{L}_{\text{RankNet}} = \frac{1}{N} \sum_{i=1}^{N} \sum_{d_j \in D} \sum_{d_k \in D} \mathbf{1}_{r_j < r_k} \log(1 + e^{(s(q_i, d_j)/\tau - s(q_i, d_k)/\tau)}), \quad (3)$$

where $D$ is the same set of documents as used in contrastive learning (including both positive and negative). $\tau$ is set to 0.1 in RankNet loss to scale the similarity score. $r_j$ is the rank of document $d_j$ among $D$, and the smaller the rank, the more relevant. For example, $r_j = 2$ means $d_j$ ranks second among $|D|$ documents. Following (Sun et al., 2023; Pradeep et al., 2023), we can leverage a powerful LLM to generate the full ranking permutation and obtain a set of pairwise relative relevance orders. The final training objective of stage II combines retrieval and reranking losses:

$$\mathcal{L} = \mathcal{L}_{\text{InfoNCE}} + \lambda \mathcal{L}_{\text{RankNet}}, \quad (4)$$

where $\lambda$ is a hyperparameter that balances two tasks, which is set to 2.0 based on prior experiments.

# 4 EXPERIMENTS

## 4.1 EXPERIMENTAL SETUP

**Base LLMs** We conduct our main experiments with open-weight, instruction-tuned LLMs from the Qwen3 family (Yang et al., 2025) across different sizes, including 0.6B, 4B, and 8B.

**Training Datasets** At Stage I, we use the public portion of the E5 training dataset (Wang et al., 2023) with roughly 1.5 million samples, curated by Springer et al. (2025). For the second stage training, we compare two different training datasets. First, we use the MS MARCO training data provided by Pradeep et al. (2023) with 40K samples. The second training dataset we use is a mixture, consisting of some of the retrieval datasets from the Stage I, as well as 2 additional public Chinese retrieval datasets from BGE-M3 training dataset (Chen et al., 2024a). We further sample instances from these datasets and construct hard negatives for each query, resulting in about 87k training samples each with 1 query, 1 positive, and 15 negatives. We also leverage Qwen3-32B for labeling the ranking permutation. For more details about the datasets, please refer to Appendix B.

**Implementation Details** We train the embedding model with full parameters for 1 epoch with a batch size of 512, using a learning rate of 5e-6. At the second stage, we continue to train the model for about 700 steps with a batch size of 128, and the number of negatives is 15. We provide other hyperparameters in Appendix C.

## 4.2 RERANKING PERFORMANCE

**Datasets** Following Sun et al. (2023), we use TREC DL dataset (Craswell et al., 2020) and a subset of BEIR (Thakur et al., 2021) for evaluation of general reranking ability. Specifically, we conduct evaluations on 8 datasets of BEIR that contain a relatively small number of queries, including TREC Covid, NFCorpus, Touch2020, DBPedia, SciFact, Signal1M, TREC News, and Robust04. Since the

Table 1: Performance comparison on TREC DL and BEIR benchmarks across LLMs. We **bold** the best performance for each task with each base LLM. The second column denotes the training data.

| Model | Data | DL19 | DL20 | Coivd | NFCorpus | Touche | DBPedia | SciFact | Signal | News | Robust | Avg. |
|-------|------|------|------|-------|----------|--------|---------|---------|--------|------|--------|------|
| BM25 | - | 50.58 | 47.96 | 59.47 | 30.75 | 44.22 | 31.80 | 67.89 | 33.05 | 39.52 | 40.70 | 43.43 |
| RankQwen3-0.6B | MS. | 69.11 | 67.74 | 78.35 | 36.41 | 37.54 | 39.19 | 71.01 | 30.96 | 44.43 | 46.31 | 48.03 |
| E$^2$RANK-0.6B | MS. | 70.78 | **70.55** | **80.03** | 37.63 | 36.60 | 41.90 | 73.19 | **35.66** | 51.17 | 49.70 | 50.74 |
| | | +1.67 | +2.81 | +1.68 | +1.22 | -0.94 | +2.71 | +2.18 | +4.70 | +6.74 | +3.39 | +2.71 |
| E$^2$RANK-0.6B | Mixed | **70.84** | 70.15 | 79.17 | **38.60** | **41.91** | **41.96** | **73.43** | 35.26 | **52.75** | **53.67** | **52.09** |
| | | +1.73 | +2.41 | +0.82 | +2.19 | +4.37 | +2.77 | +2.42 | +4.30 | +8.32 | +7.36 | +4.07 |
| RankQwen3-4B | MS. | **72.36** | 69.83 | 83.91 | **39.88** | 32.66 | **43.91** | 76.37 | 32.15 | 50.81 | 59.36 | 52.38 |
| E$^2$RANK-4B | MS. | 70.67 | **71.05** | **84.90** | 39.32 | 35.44 | 43.66 | **77.69** | 34.21 | 51.22 | 57.49 | 52.99 |
| | | -1.69 | +1.22 | +0.99 | -0.56 | +2.78 | -0.25 | +1.32 | +2.06 | +0.41 | -1.87 | +0.61 |
| E$^2$RANK-4B | Mixed | 70.44 | 70.64 | 83.30 | 39.20 | **43.16** | 42.95 | 77.19 | **34.48** | 52.71 | **60.16** | **54.14** |
| | | -1.92 | +0.81 | -0.61 | -0.68 | +10.50 | -0.96 | +0.82 | +2.33 | +1.90 | +0.80 | +1.76 |
| RankQwen3-8B | MS. | **73.15** | 70.75 | 85.37 | **40.05** | 31.73 | **45.44** | 78.96 | 32.48 | 52.36 | **60.72** | 53.39 |
| E$^2$RANK-8B | MS. | 71.66 | 70.87 | **85.43** | 39.57 | 36.59 | 44.26 | 78.17 | 33.52 | **55.36** | 57.95 | 53.86 |
| | | -1.49 | +0.12 | +0.06 | -0.48 | +4.86 | -1.18 | -0.79 | +1.04 | +3.00 | -2.77 | +0.47 |
| E$^2$RANK-8B | Mixed | 72.95 | **71.16** | 84.09 | 39.08 | **42.06** | 43.44 | 77.49 | **34.01** | 54.25 | 60.34 | **54.35** |
| | | -0.20 | +0.41 | -1.28 | -0.97 | +10.33 | -2.00 | -1.47 | +1.53 | +1.89 | -0.38 | +0.96 |

rise of reasoning-intensive ranking for complex retrieval-augmented tasks like DeepResearch, we also evaluate E$^2$RANK on BRIGHT (Su et al., 2025). We use BM25 as the first-stage retriever for TREC DL and BEIR and use ReasonIR (Shao et al., 2025) with GPT4 reason-query for BRIGHT. For all benchmarks, we rerank the top-100 candidate documents and use NDCG@10 as the metric.

**Baselines** In order to achieve a fair and direct comparison, we used the same base LLM to compare RankGPT-like listwise rerank with E$^2$RANK, and finetune Qwen3 on the training data provided by Pradeep et al. (2023), denoted as RankQwen3. More training details will be provided in Appendix C. For RankQwen3, we use a sliding window strategy of window size 20 and step 10; while for our model, we only feed the top-20 documents to the listwise prompt and use its embedding to rerank the top-100. We believe that this direct comparison between E$^2$RANK and RankQwen3 without the influence of base LLMs and training data can provide richer insights.

For reference, we also report other baseline results on TREC DL and BEIR, including cross-encoders monoBERT (Nogueira et al., 2019), monoT5 (Nogueira et al., 2020), and RankT5 (Zhuang et al., 2023), as well as listwise LLM-based rerankers ListT5 (Yoon et al., 2024), RankZephyr (Pradeep et al., 2023), and RankGPT (Sun et al., 2023). As for the baselines of BRIGHT, we compare E$^2$RANK with reasoning rerankers with parameters less than 14B, including Rank-R1 (Zhuang et al., 2025), Rank1 (Weller et al., 2025), JudgeRank (Niu et al., 2024), Rearank (Zhang et al., 2025a), ERank (Cai et al., 2025), and ReasonRank (Liu et al., 2025c). Note that only RankGPT and JudgeRank are zero-shot; others are all fine-tuned, and most reasoning rerankers are trained with RL.

**E$^2$RANK consistently outperforms RankQwen3.** We present the direct comparison between E$^2$RANK and RankQwen3 on general reranking datasets in Table 1. When training on the same datasets, our proposed E$^2$RANK demonstrates a clear and consistent advantage over RankQwen3 baseline across all model sizes, especially for the 0.6B model with an average gain of +2.71 NDCG@10, while E$^2$RANK-4B and E$^2$RANK-8B show smaller but stable improvements on average. Additionally, using more diverse and richer datasets can further boost the performance of E$^2$RANK. As model size grows, both RankQwen3 and E$^2$RANK improve over BM25, but E$^2$RANK-8B trained on mixed datasets achieves the best overall performance.

**E$^2$RANK achieves competitive rerank accuracy across other strong baselines.** Table 2 presents broader comparisons on the TREC DL and BEIR benchmarks, and our models compete effectively with a diverse array of state-of-the-art rerankers. Compared to fine-tuned pointwise rerankers such as monoBERT and monoT5, our approach achieves significantly higher average scores, and even surpasses strong listwise baselines like RankZephyr and ListT5 on BEIR benchmarks. Notably, while RankGPT-4o remains the strongest zero-shot model, our fine-tuned 8B model secures the top performance on the DL20 dataset (71.16) and achieves the highest overall BEIR average (54.35), surpassing even much larger zero-shot models like RankGPT-4o and establishing our approach as a powerful and efficient alternative to existing fine-tuned and zero-shot methods.

Table 3: Performance comparison on BRIGHT benchmarks across LLMs. We **bold** the best performance for each task and underline the second best.

| Model | StackExchange | | | | | | | Coding | | Theorem-based | | | Avg. |
| --- | --- | --- | --- | --- | --- | --- | --- | --- | --- | --- | --- | --- | --- |
| | Bio. | Econ. | Earth. | Psy. | Rob. | Stack. | Sus. | Pony. | LC. | AoPS | TheoQ. | ThoT. | |
| ReasonIR | 43.5 | 32.8 | 43.0 | 38.9 | 21.1 | 30.6 | 27.3 | 31.6 | 19.6 | 7.3 | 36.7 | 34.1 | 30.5 |
| RankT5 (3B) | 11.4 | 22.1 | 10.9 | 13.6 | 11.4 | 11.4 | 16.0 | 27.5 | 38.1 | 9.2 | 18.3 | 9.5 | 16.6 |
| RankZephyr | 19.9 | 17.4 | 12.4 | 34.9 | 24.7 | 13.4 | 22.3 | 29.3 | 32.4 | 6.1 | 29.0 | 30.1 | 22.6 |
| Rank-R1 (7B) | 39.3 | 28.1 | 23.9 | 30.0 | 17.3 | 18.1 | 33.2 | 18.6 | 15.0 | 4.2 | 25.4 | 35.7 | 24.1 |
| Rank-R1 (14B) | 27.4 | 38.7 | 23.1 | 44.5 | 37.1 | 27.8 | 36.8 | 21.3 | 19.2 | 8.8 | 31.7 | 39.5 | 29.7 |
| Rank1 (7B) | 44.1 | 33.5 | 21.8 | 30.0 | 15.0 | 22.1 | 28.5 | 11.8 | 21.7 | 1.2 | 26.2 | 36.2 | 24.3 |
| Rearank (7B) | 35.3 | 29.8 | 25.5 | 35.7 | 19.1 | 20.1 | 32.9 | 29.9 | 20.2 | 6.2 | 36.7 | 38.3 | 27.5 |
| JudgeRank (8B) | 37.1 | 27.2 | 19.2 | 28.6 | 11.6 | 19.9 | 22.5 | 10.2 | 10.2 | 3.6 | 22.9 | 29.4 | 20.2 |
| ERank (4B) | 42.1 | 42.5 | 26.3 | 36.4 | 20.8 | 27.3 | 33.2 | 31.7 | 21.8 | 10.9 | 32.8 | 40.6 | 30.5 |
| ERank (14B) | 46.6 | 42.5 | 25.2 | 37.3 | 19.6 | 30.2 | 34.6 | **31.9** | 25.6 | 10.5 | 32.4 | **45.0** | 31.8 |
| ReasonRank (7B) | 35.1 | 47.8 | 31.2 | **56.7** | **47.8** | 32.5 | **40.9** | 23.2 | 25.0 | 7.7 | 39.5 | 41.8 | **35.7** |
| RankQwen3-0.6B | 44.7 | 38.7 | 28.4 | 40.4 | 20.5 | 26.1 | 28.5 | 19.9 | 29.1 | 6.8 | 35.8 | 30.5 | 29.1 |
| E$^2$RANK-0.6B (MS.) | 41.2 | 46.5 | 30.9 | 34.3 | 24.2 | 24.0 | 27.4 | 9.4 | 35.6 | 10.5 | 30.4 | 27.2 | 28.5 |
| E$^2$RANK-0.6B | 44.1 | 46.5 | 31.0 | 40.8 | 26.1 | 30.6 | 30.6 | 11.7 | **38.5** | 8.0 | 35.9 | 28.0 | 31.0 |
| RankQwen3-4B | 47.0 | 44.2 | 25.2 | 44.7 | 24.1 | 29.7 | 41.1 | 22.6 | 22.0 | 9.0 | 38.2 | 36.0 | 32.0 |
| E$^2$RANK-4B (MS.) | 42.9 | 44.4 | 30.5 | 39.1 | 27.4 | 25.4 | 32.9 | 7.7 | 38.3 | **11.4** | 38.4 | 35.5 | 31.2 |
| E$^2$RANK-4B | 47.6 | 46.7 | 31.8 | 43.1 | 26.8 | 31.4 | 34.6 | 8.6 | 38.4 | 8.2 | **39.8** | 31.6 | 32.4 |
| RankQwen3-8B | **49.5** | 44.2 | 30.4 | 44.9 | 24.9 | 26.1 | 39.6 | 18.8 | 20.8 | 7.6 | 39.0 | 37.9 | 32.0 |
| E$^2$RANK-8B (MS.) | 45.0 | **48.1** | 31.9 | 38.5 | 29.3 | 32.2 | 36.4 | 10.3 | 36.0 | 10.6 | 37.9 | 36.6 | 32.7 |
| E$^2$RANK-8B | 49.2 | 47.2 | **32.3** | 44.7 | 28.2 | **32.9** | 38.4 | 10.6 | 36.2 | 8.2 | 38.2 | 33.4 | 33.4 |

**Efficiency Analysis.** We conduct the efficiency analysis on the Covid dataset using a single NVIDIA A100 80G GPU. The Covid dataset contains 50 test queries, and the average length of documents tokenized by the Qwen3 tokenizer is approximately 350. We implement the evaluation code using vLLM (Kwon et al., 2023), a highly-efficient LLM inference infrastructure. As shown in Figure 1 (b), E$^2$RANK significantly reduces inference latency across all model sizes compared to RankQwen3, achieving up to about $5\times$ speedup at 8B while maintaining superior ranking performance. Even E$^2$RANK-8B model is faster than RankQwen3-0.6B. Since RankQwen3 uses a sliding window strategy, it can't use the batch inference techniques for inference, while full ranking is less effective. In contrast, E$^2$RANK inherits the advantages of the embedding model, supports batch inference, and can encode document embeddings offline, further reducing online reranking latency. The detailed results of reranking latency are listed in Appendix G, Table 12 and 13.

Table 2: Performance comparison across broader baselines. The best result of each benchmark is **bolded**, and the second best is underlined.

| Model | DL19 | DL20 | BEIR Avg. |
| --- | --- | --- | --- |
| BM25 | 50.58 | 47.96 | 43.43 |
| *Fine-tuned Pointwise Reranker* | | | |
| monoBERT (340M) | 70.50 | 67.28 | 47.16 |
| monoT5 (3B) | 71.83 | 68.89 | 51.36 |
| RankT5 (3B) | 72.50 | 70.40 | 52.50 |
| *Fine-tuned Listwise Reranker* | | | |
| ListT5 (3B) | 71.80 | 69.10 | 53.00 |
| RankZephyr | 73.39 | 70.02 | 51.15 |
| *Zero-shot Listwise Reranker* | | | |
| RankQwen3 (14B) | 74.19 | 69.10 | 53.67 |
| RankGPT-4o | **74.78** | 69.52 | 53.09 |
| RankGPT-4o-mini | 72.36 | 67.30 | 51.16 |
| *Ours* | | | |
| E$^2$RANK-0.6B | 70.84 | 70.15 | 52.09 |
| E$^2$RANK-4B | 70.44 | 70.64 | 54.14 |
| E$^2$RANK-8B | 72.95 | **71.16** | **54.35** |

**E$^2$RANK demonstrates strong performance on the BRIGHT benchmark.** On the challenging BRIGHT benchmark, E$^2$RANK delivers robust performance, as shown in Table 3. Without any RL or reasoning process, E$^2$RANK-8B attains a highly competitive average score of 33.4, surpassing RankQwen3 and most reasoning rerankers and only underperforming ReasonRank trained on synthetic reasoning data, validating the strong generalization capabilities.

## 4.3 EMBEDDING ABILITY

**Benchmark and Baselines** We evaluate E$^2$RANK on the Massive Text Embedding Benchmark (MTEB) (Muennighoff et al., 2022). Specifically, we mainly evaluate its English v1 version, a collection of 56 datasets covering seven types of embedding tasks: classification, clustering, pairwise classification, reranking, retrieval, sentence similarity (STS), and summarization. We also leverage

Table 4: Performance comparison on MTEB. Note that some baselines are trained with non-public data, and we only report the version trained on public data, marked using *. The best results for each subtask are highlighted in **bold**, and the second-best results are underlined.

| Categorias →
# of datasets → | Retr.
15 | Rerank.
4 | Clust.
11 | PairClass.
3 | Class.
12 | STS
10 | Summ.
1 | Avg.
56 |
|---|---|---|---|---|---|---|---|---|
| Instructor-xl | 49.26 | 57.29 | 44.74 | 86.62 | 73.12 | 83.06 | **32.32** | 61.79 |
| BGE$_{\text{large-en-v1.5}}$ | 54.29 | 60.03 | 46.08 | 87.12 | 75.97 | 83.11 | 31.61 | 64.23 |
| GritLM$_{\text{Mistral-7b-v1}}$* | 53.10 | **61.30** | **48.90** | 86.90 | 77.00 | 82.80 | 29.40 | 64.70 |
| E5$_{\text{Mistral-7b-v1}}$* | 52.78 | 60.38 | 47.78 | **88.47** | 76.80 | 83.77 | 31.90 | 64.56 |
| Echo$_{\text{Mistral-7b-v1}}$ | 55.52 | 58.14 | 46.32 | 87.34 | **77.43** | 82.56 | 30.73 | 64.68 |
| LLM2Vec$_{\text{Mistral-7B}}$ | 55.99 | 58.42 | 45.54 | 87.99 | 76.63 | 84.09 | 29.96 | 64.80 |
| LLM2Vec$_{\text{Meta-LLaMA-3-8B}}$ | 56.63 | 59.68 | 46.45 | 87.80 | 75.92 | 83.58 | 30.94 | 65.01 |
| BGE-en-icl$_{\text{Mistral-7b-v1}}$* (zero-shot) | **59.59** | 56.85 | 42.61 | 87.87 | 75.47 | 83.30 | 29.52 | 64.67 |
| E$^2$RANK-0.6B (w/ only Stage I) | 48.07 | 56.16 | 42.38 | 82.47 | 72.05 | 80.90 | 29.84 | 60.05 |
| E$^2$RANK-0.6B | 51.74 | 55.97 | 40.85 | 83.93 | 73.66 | 81.41 | 30.90 | 61.25 |
| E$^2$RANK-4B (w/ only Stage I) | 54.36 | 59.30 | 44.62 | 84.36 | 76.11 | 82.31 | 29.33 | 63.61 |
| E$^2$RANK-4B | 55.33 | 59.10 | 44.27 | 87.14 | 77.08 | 84.03 | 30.06 | 64.47 |
| E$^2$RANK-8B (w/ only Stage I) | 55.31 | 55.73 | 45.84 | 85.23 | 75.69 | 83.23 | 29.66 | 64.26 |
| E$^2$RANK-8B | 56.89 | 59.58 | 44.75 | 86.96 | 76.81 | **84.52** | 30.23 | **65.03** |

Table 5: End-to-end ranking performance.

| | | DL20 | BEIR | BRIGHT |
|---|---|---|---|---|
| E$^2$RANK-0.6B | Retrieval | 66.77 | 47.60 | 18.37 |
| | + Rerank | 74.40 | 50.66 | 22.58 |
| E$^2$RANK-4B | Retrieval | 74.00 | 52.11 | 27.84 |
| | + Rerank | 76.88 | 54.12 | **32.15** |
| E$^2$RANK-8B | Retrieval | 75.83 | 53.39 | 25.09 |
| | + Rerank | **78.02** | **55.08** | 31.00 |

Table 6: Ablation on different training strategies.

| | DL20 | BEIR | BRIGHT | MTEB(v2) |
|---|---|---|---|---|
| E$^2$RANK-0.6B | **70.15** | 52.09 | **30.96** | 63.41 |
| w/o Stage I | 69.32 | 51.33 | 30.66 | 60.61 |
| w/o InfoNCE in Stage II | 69.11 | **52.17** | 29.99 | 61.92 |
| w/ only Stage I | 63.55 | 46.31 | 15.30 | 62.40 |
| w/o RankNet in Stage II | 66.50 | 49.24 | 22.40 | 63.31 |
| w/o Listwise in Stage II | 66.29 | 49.93 | 22.69 | **63.66** |

its English v2 version for quick evaluation and ablation studies, which is smaller and cleaner with 41 tasks. We compare our models with recent advanced open source text embedding models that are trained on public datasets, including Instructor-xl (Su et al., 2022), BGE-large-en-v1.5 (Xiao et al., 2023), GritLM (Muennighoff et al., 2024), E5 (Wang et al., 2023), EchoEmbedding (Springer et al., 2025), LLM2Vec (BehnamGhader et al., 2024), and BGE-ICL (Li et al., 2024a).

**Results** Table 4 presents the performance of E$^2$RANK on the MTEB(Eng, v1) benchmark. When leveraging only the public dataset, E$^2$RANK demonstrates strong embedding capabilities, while E$^2$RANK-8B shows slight performance advantages on average compared to previous advanced models. Notably, compared with the variant with only contrastive learning, distilling from richer ranking signals will bring consistent and significant enhancements in retrieval tasks (↑ 1.58 for E$^2$RANK-8B), demonstrating the effectiveness of the ranking objective. Noticed that here we focus on general and pure embedding ability, so we do not use the listwise prompt for reranking tasks.

### 4.4 UNIFIED AND END-TO-END RETRIEVAL AND RERANKING

We also perform end-to-end ranking to evaluate if the single E$^2$RANK model could be a unified model in the search paradigm. Specifically, we use E$^2$RANK first to retrieve the top-100 candidate documents and then use it to rerank these documents further.

The results in Table 5 indicate that using a single E$^2$RANK model for both retrieval and reranking leads to consistent improvements across different model scales and datasets. Notably, as the model size increases from 0.6B to 8B parameters, we observe progressive gains in end-to-end ranking performance on all benchmarks. Additionally, reranking consistently enhances the initial retrieval performance, with the E$^2$RANK-8B achieving the best performance of 55.08 nDCG@10 on BEIR after reranking. These results demonstrate the viability of using a single unified model for both stages of the search pipeline, thereby reducing system complexity and latency while maintaining strong performance.

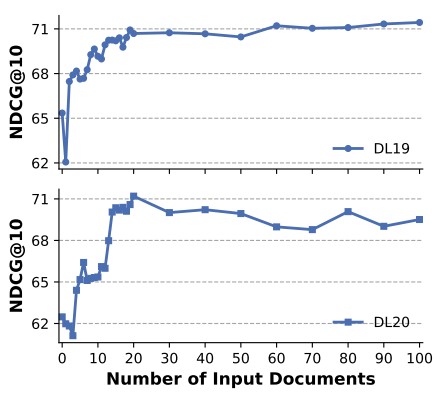

Figure 2: Trend of NDCG@10 changes with the number of input documents in listwise prompt.

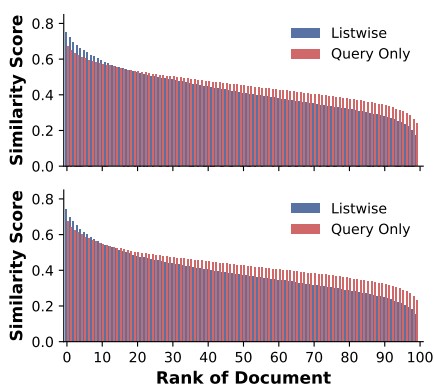

Figure 3: Score distribution of using listwise (with 20 documents) and non-listwise prompts.

## 4.5 ABLATION STUDY

We evaluate the effectiveness of different training strategies and conduct ablation studies using the Qwen3-0.6B model on TREC DL20, BEIR, BRIGHT, and MTEB(eng, v2). The reranking settings and metrics are the same as in Section 4.2. The results shown in Table 6 indicate that the full training strategy achieves the best or highly competitive performance across all datasets, demonstrating the effectiveness of the integrated design. For the last three lines, we use query-only embedding instead of listwise prompt for evaluation since they are not trained on it. We provide more analysis in the Appendix.

**The first-stage contrastive learning is crucial for foundational query-document alignment and embedding ability.** Its removal causes consistent performance degradation, especially on MTEB. This confirms that initial large-scale contrastive learning provides an essential foundation for subsequent ranking tasks.

**The RankNet loss is the most critical element for effective ranking.** Removing the RankNet loss causes the most severe performance collapse, particularly on BEIR and BRIGHT. This underscores that the pairwise ranking objective is indispensable for learning complex relevance ordering patterns.

**The listwise prompts incorporating PRF documents substantially enhance ranking effectiveness.** If retaining the RankNet loss but removing the listwise prompt, the ranking performance will still be greatly affected (last line). This indicates that the reranking ability is mainly from the listwise prompt with PRF signals, but not the richer training labels.

## 4.6 ANALYSIS

In order to understand the reranking behaviors of $E^2$RANK, we conduct a further analysis.

**Comparison with other PRF methods.** For comparison, we implemented two classical PRF-style baselines using the model without listwise training (corresponding to "w/o Listwise in Stage II" in Table 6), and applied (i) a text-based listwise prompt, and (ii) a vector-based Rocchio-style fusion of query and document embeddings. The results are shown in Table 7. While both PRF baselines introduce PRF-like signals, neither comes close to the performance of $E^2$RANK, and in some cases even harms ranking quality. This demonstrates that simply injecting PRF signals is insufficient and the model must be trained

Table 7: Performance comparison across other PRF baselines. The best result of each benchmark is **bolded**.

| | DL20 | BEIR | BRIGHT |
|---|---|---|---|
| $E^2$RANK-0.6B | **70.15** | **52.09** | **30.96** |
|   w/o Listwise Prompt | 65.25 | 49.46 | 21.50 |
| w/o Listwise in Stage II | 66.29 | 49.93 | 22.69 |
|   + text-based PRF | 56.57 | 46.52 | 29.62 |
|   + vector-based PRF | 63.96 | 49.20 | 21.85 |

to understand how to use these signals. In contrast, rather than just enriching the query, $E^2$RANK learns how to perform ranking-aware feature transformation conditioned on a set of candidate documents, which requires supervision to learn head–tail interactions, semantic reinforcement, and conflict resolution among top candidates. More details and discussion could be found in Appendix D.

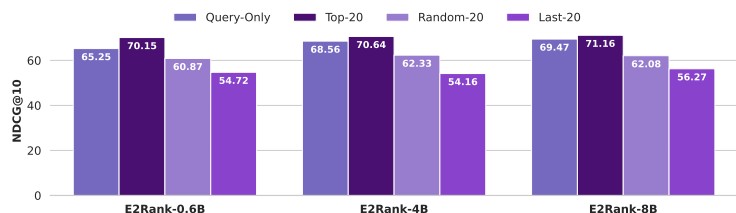

Figure 4: Results of selecting different documents as PRF on DL20.

**Influence of number of input documents in the listwise prompts.** Figure 2 shows that when the number of input documents is small (less than 20), incorporating more documents into the list-wise prompt consistently improves ranking performance. This trend can be interpreted as additional documents enriching the query with pseudo-relevance signals, allowing the model to capture the fine-grained relevance. Notably, the gains plateau after around 20 documents, indicating that the marginal benefit of adding more feedback signals diminishes once the prompt already captures sufficient relevance context, and may even bring negative benefits on different datasets.

**Influence of the selection of documents.** To evaluate the impact of document selection within the listwise prompt, we conducted an ablation study varying the composition of documents in four settings, as presented in Figure 4. Across all model sizes, using the Top-20 retrieved documents consistently yielded the best ranking performance. In contrast, Random-20 and Last-20 settings significantly reduced performance, even falling below the query-only baseline. This shows that the benefit of document context comes not from adding more text, but from leveraging highly relevant documents, which is just like the working mechanism of PRF. Conversely, low-quality or off-topic documents introduce noise and conflicting signals, leading to deterioration in ranking quality.

**Similarity score distribution.** Figure 3 further analyzes how this pseudo relevance feedback affects the ranking behavior by comparing similarity score distributions between listwise and non-listwise settings. Specifically, we sort the reranking scores of 100 documents from high to low, and take the average of all queries for the rank position. We can see that the listwise prompts yield consistently higher similarity scores for top-ranked documents while maintaining a steeper decline for lower-ranked ones, suggesting sharper discrimination between relevant and irrelevant documents. In contrast, the query-only setting produces a flatter score distribution. This demonstrates that listwise prompts with PRF enhance $E^2$RANK 's ability to allocate higher scores to truly relevant documents.

**Influence of different first-stage retrievers.** We evaluate the $E^2$RANK's reranking ability under different first-stage retrievers, and detail the results in Appendix G, Table 27. Across all retrievers, $E^2$RANK consistently improves the performance, demonstrating its generalization ability and robustness while adapting to varying initial retrieval qualities as a reranker. Additionally, this also indicates that better search results as better PRF can lead to better ranking performance.

## 5 CONCLUSION

In this paper, we propose $E^2$RANK, a unified framework that enables a single text embedding model to perform both efficient retrieval and high-quality listwise reranking, by reformulating the listwise reranking prompt as a pseudo relevance feedback query. Extensive experiments demonstrate that $E^2$RANK can be an independent reranker and achieve state-of-the-art reranking performance on BEIR and strong results on BRIGHT, while significantly reducing inference latency compared to existing RankGPT-like listwise rerankers. Moreover, $E^2$RANK maintains competitive embedding capabilities on the MTEB benchmark. Our work highlights the potential of single embedding models to serve as unified retrieval-reranking engines, offering a practical, efficient, and accurate alternative to complex multi-stage ranking systems.

## ETHICS AND REPRODUCIBILITY STATEMENT

This study does not raise concerns related to discrimination, bias, or fairness. To ensure repro-ducibility, we provide detailed descriptions of the experimental setup in Section 4.1 and additional

implementation details in Appendix C. All data used in our experiments are obtained from previously released and widely adopted datasets. with details in Appendix B. All open source libraries and resources used in this study are also fully specified. We also provide the complete source code for reproduction directly in the supplementary material.

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

## A  THE USE OF LARGE LANGUAGE MODELS

We only used large language models (LLMs) as auxiliary tools for grammar checking, language polishing, and logo generation. All outputs were carefully reviewed by the authors, who take full responsibility for the final manuscript.

## B  TRAINING DATASET DETAILS

**Dataset composition**  We mainly leverage the public portion of the E5 dataset (Wang et al., 2023). Specifically, for the training at Stage I, we use the sampled version with around 1.5 million samples in total, which is constructed by Springer et al. (2025) and is also used by LLM2Vec (BehnamGhader et al., 2024). The mixture consists of ELI5, HotpotQA, FEVER, MIRACL, MS MARCO passage ranking and document ranking, NQ, NLI, SQuAD, TriviaQA, Quora Duplicate Questions, Mr.TyDi, DuReader, and T2Ranking. Each query in the datasets has only one positive and one negative.

As for the training at Stage II, since we need more negatives to meet the training objective, the E5 dataset cannot fully meet our requirements. Therefore, we used the dataset from BGE-M3 (Chen et al., 2024a), where each query contains multiple negatives. Specifically, we mainly used the retrieval dataset from the intersection of the E5 dataset and BGE-M3 dataset, including HotpotQA, MIRACL, MSMARCO passage, NQ, TriviaQA, DuReader, and T2Ranking. In addition, we have added two widely used Chinese retrieval datasets, cMedQAv2 and MMarco Chinese, which are included in the BGE-M3 dataset. Due to the length division of the BGE-M3 dataset, we only used the parts with document lengths less than 500. Meanwhile, we filtered queries containing fewer than 15 negative examples and further downsampled the dataset. In the end, we obtained a mixed dataset containing approximately 157k samples, with each instance containing one query, one negative, and fifteen negatives.

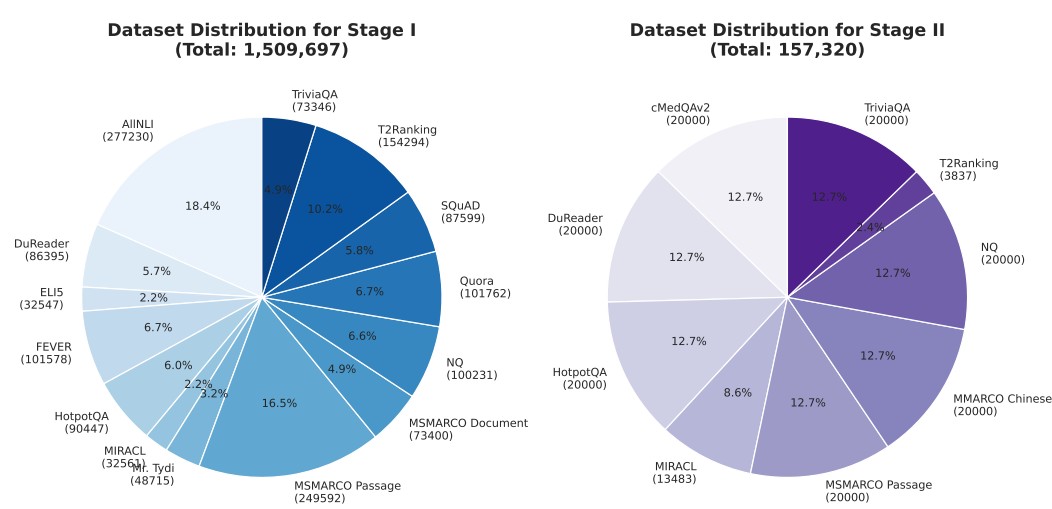

Figure 5: Dataset distribution for training.

**Producing full ranking labels using Qwen3-32B**  We leverage Qwen3-32B (disabled thinking mode) (Yang et al., 2025) to generate the full ranking labels for the Stage II training data. The process is similar to RankZephyr's (Pradeep et al., 2023). Specifically, we use the instruction in Table 9 to have the model generate a ranking list in text form, and then parse the text. Then, we filter the results with the wrong output formations, which is only a very small portion of the entire dataset. The instruction used for each dataset is adapted from BehnamGhader et al. (2024), which can be found in Table 8.

Interestingly, we calculate the "accuracy" of model annotation, which refers to the frequency at which the model places the "golden positive" in the dataset at the top of its ranking. The results are shown in Figure 6. We can see that the LLM's judgment and actual annotation of the most relevant

Table 8: Instructions used for each of the E5 datasets in Stage I.

| Dataset | Instruction(s) |
|---|---|
| NLI | Given a premise, retrieve a hypothesis that is entailed by the premise |
| | Retrieve semantically similar text |
| DuReader | Given a Chinese search query, retrieve web passages that answer the question |
| ELI5 | Provided a user question, retrieve the highest voted answers on Reddit ELI5 forum |
| FEVER | Given a claim, retrieve documents that support or refute the claim |
| HotpotQA | Given a multi-hop question, retrieve documents that can help answer the question |
| MIRACL | Given a question, retrieve Wikipedia passages that answer the question |
| MrTyDi | Given a question, retrieve Wikipedia passages that answer the question |
| MSMARCO Passage | Given a web search query, retrieve relevant passages that answer the query |
| MSMARCO Document | Given a web search query, retrieve relevant documents that answer the query |
| NQ | Given a question, retrieve Wikipedia passages that answer the question |
| QuoraDuplicates | Given a question, retrieve questions that are semantically equivalent to the given question |
| | Find questions that have the same meaning as the input question |
| SQuAD | Retrieve Wikipedia passages that answer the question |
| T2Ranking | Given a Chinese search query, retrieve web passages that answer the question |
| TriviaQA | Retrieve Wikipedia passages that answer the question |

documents are not always consistent. Especially in the MS MARCO dataset, the consistency rate only barely exceeds half. Previous work discussed and compared using the golden label and using a reranker for labeling, but they didn't leverage LLMs (Zhang et al., 2023b). Since the construction of the dataset is not the focus of this article, we will not discuss this discovery in detail and will leave higher-quality dataset construction schemes for future work.

Table 9: Instruction for generating full ranking labels.

```
<|im_start|>user
I will provide you with {N} passages, each indicated by
a numerical identifier []. Rank the passages based on
their relevance to the search query: {query}.
Documents:
[1] {document 1}
[2] {document 2}
...
[N] {document N}
Search Query: {query}
Rank the {N} passages above based on their relevance to
the search query. All the passages should be included
and listed using identifiers, in descending order of
relevance. The output format should be [] > [] > ...,
e.g., [4] > [2] > ..., Only respond with the ranking
results, do not say anything else or explain.  <|im_end|>
<|im_start|>assistant
<think>\n\n</think>\n\n
```

## C   IMPLEMENTATION DETAILS

In this section, we provide a detailed introduction to our training settings.

**Stage I training**   All models are trained with full parameters, DeepSpeed Zero3, brain floating point (BF16) quantization, and gradient checkpointing to optimize GPU memory consumption. We train on 8 NVIDIA A100 80G GPUs with an effective batch size of 512 for 1 epoch using a maximum sequence length of 512 tokens. We use a learning rate of $2 \times 10^{-5}$ and a linear learning rate warm-up for the first 300 steps.

**Stage II training**   For the training data, it is important to note that we do not use the full datasets introduced in Appendix B for training. Instead, for each dataset, we only sample at most 10,000 instances, leading to around 87k training instances actually.

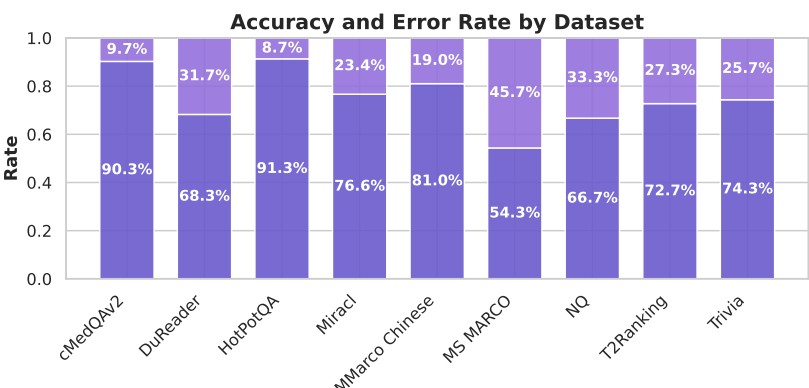

Figure 6: Accuracy for labeling the datasets.

Table 10: Instructions in listwise prompts used for each of the datasets in Stage II.

| Dataset | Instruction(s) |
|---|---|
| DuReader | Given a Chinese search query and some relevant documents, rerank the documents that answer the query |
| HotpotQA | Given a multi-hop question and some relevant documents, rerank the documents that answer the question |
| MIRACL | Given a question and some relevant Wikipedia documents, rerank the documents that answer the question |
| MSMARCO Passage | Given a web search query and some relevant documents, rerank the documents that answer the query |
| NQ | Given a question, retrieve Wikipedia passages that answer the question |
| T2Ranking | Given a Chinese search query and some relevant documents, rerank the documents that answer the query |
| TriviaQA | Given a question and some relevant Wikipedia documents, rerank the documents that answer the question |

The instructions for the listwise prompt are listed in Table 10.

We train all models on 8 NVIDIA A100 80G GPUs with an effective batch size of 128 for 1 epoch (each instance contains multiple documents). We use DeepSpeed Zero3, BF16, and gradient check-pointing to optimize GPU memory consumption. For documents, we use a maximum length of 1024. We also use in-batch negatives. We use a learning rate initialized at $5 \times 10^{-6}$ with a linear scheduler and a warmup ratio of 0.03.

**Training RankQwen3**  We fine-tune the Qwen3 model on the GPT-4 labeled listwise ranking dataset provided by Pradeep et al. (2023). The dataset contains 40k samples, and we train the model for 1 epoch with a batch size of 16 per device, leading to an effective batch size of 64. For different sizes, we adjust the gradient accumulation steps to fit the batch size. We use DeepSpeed and BF16 mixed precision for acceleration. The learning rate is initialized at $5 \times 10^{-6}$ with a linear scheduler and a warmup ratio of 0.03. The training is performed on 4 NVIDIA A100 80G GPUs. We use LLM4Ranking Framework (Liu et al., 2025a) for training and evaluation.

**Evaluation details**  We use the following instruction for the evaluation of all reranking tasks:

```
<|im_start|>user
Given a web search query and some relevant documents,
rerank the documents that answer the query:
Documents:
[1] {document 1}
[2] {document 2}
...
[N] {document N}
Search Query:  {query}
<|im_end|>
<|im_start|>assistant
<think>\n\n</think>\n\n
```

Table 11: Instructions used for evaluation on the MTEB benchmark. "STS*" refers to all the STS tasks.

| Task Name | Instruction |
|---|---|
| AmazonCounterfactualClassif. | Classify a given Amazon customer review text as either counterfactual or not-counterfactual |
| AmazonPolarityClassification | Classify Amazon reviews into positive or negative sentiment |
| AmazonReviewsClassification | Classify the given Amazon review into its appropriate rating category |
| Banking77Classification | Given a online banking query, find the corresponding intents |
| EmotionClassification | Classify the emotion expressed in the given Twitter message into one of the six emotions: anger, fear, joy, love, sadness, and surprise |
| ImdbClassification | Classify the sentiment expressed in the given movie review text from the IMDB dataset |
| MassiveIntentClassification | Given a user utterance as query, find the user intents |
| MassiveScenarioClassification | Given a user utterance as query, find the user scenarios |
| MTOPDomainClassification | Classify the intent domain of the given utterance in task-oriented conversation |
| MTOPIntentClassification | Classify the intent of the given utterance in task-oriented conversation |
| ToxicConversationsClassif. | Classify the given comments as either toxic or not toxic |
| TweetSentimentClassification | Classify the sentiment of a given tweet as either positive, negative, or neutral |
| ArxivClusteringP2P | Identify the main and secondary category of Arxiv papers based on the titles and abstracts |
| ArxivClusteringS2S | Identify the main and secondary category of Arxiv papers based on the titles |
| BiorxivClusteringP2P | Identify the main category of Biorxiv papers based on the titles and abstracts |
| BiorxivClusteringS2S | Identify the main category of Biorxiv papers based on the titles |
| MedrxivClusteringP2P | Identify the main category of Medrxiv papers based on the titles and abstracts |
| MedrxivClusteringS2S | Identify the main category of Medrxiv papers based on the titles |
| RedditClustering | Identify the topic or theme of Reddit posts based on the titles |
| RedditClusteringP2P | Identify the topic or theme of Reddit posts based on the titles and posts |
| StackExchangeClustering | Identify the topic or theme of StackExchange posts based on the titles |
| StackExchangeClusteringP2P | Identify the topic or theme of StackExchange posts based on the given paragraphs |
| TwentyNewsgroupsClustering | Identify the topic or theme of the given news articles |
| SprintDuplicateQuestions | Retrieve duplicate questions from Sprint forum |
| TwitterSemEval2015 | Retrieve tweets that are semantically similar to the given tweet |
| TwitterURLCorpus | Retrieve tweets that are semantically similar to the given tweet |
| AskUbuntuDupQuestions | Retrieve duplicate questions from AskUbuntu forum |
| MindSmallReranking | Retrieve relevant news articles based on user browsing history |
| SciDocsRR | Given a title of a scientific paper, retrieve the titles of other relevant papers |
| StackOverflowDupQuestions | Retrieve duplicate questions from StackOverflow forum |
| ArguAna | Given a claim, find documents that refute the claim |
| ClimateFEVER | Given a claim about climate change, retrieve documents that support or refute the claim |
| CQADupstackRetrieval | Given a question, retrieve detailed question descriptions from Stackexchange that are duplicates to the given question |
| DBPedia | Given a query, retrieve relevant entity descriptions from DBPedia |
| FEVER | Given a claim, retrieve documents that support or refute the claim |
| FiQA2018 | Given a financial question, retrieve user replies that best answer the question |
| HotpotQA | Given a multi-hop question, retrieve documents that can help answer the question |
| MSMARCO | Given a web search query, retrieve relevant passages that answer the query |
| NFCorpus | Given a question, retrieve relevant documents that best answer the question |
| NQ | Given a question, retrieve Wikipedia passages that answer the question |
| QuoraRetrieval | Given a question, retrieve questions that are semantically equivalent to the given question |
| SCIDOCS | Given a scientific paper title, retrieve paper abstracts that are cited by the given paper |
| SciFact | Given a scientific claim, retrieve documents that support or refute the claim |
| Touche2020 | Given a question, retrieve detailed and persuasive arguments that answer the question |
| TRECCOVID | Given a query on COVID-19, retrieve documents that answer the query |
| STS* | Retrieve semantically similar text. |
| SummEval | Given a news summary, retrieve other semantically similar summaries |

In fact, based on our experiments, different instructions have a very small impact on performance, at least not statistically significant. So this will not affect the experimental results of the paper.

**Instructions used for evaluation of MTEB**   When evaluating MTEB, we use the same instructions as Zhang et al. (2025c). The list of instructions for each task is listed in Table 11.

# D    DISCUSSION OF PRF-LIKE MECHANISM

Pseudo-Relevance Feedback (PRF) has long been used in classical IR systems to refine the query representation using top-ranked documents, based on the assumption that these documents contain valuable signals that reflect the underlying relevance intent. Traditional PRF methods typically fall into two categories: (1) text-based PRF, where top-retrieved documents are used to extract expansion terms or passages to enrich the query (e.g., Rocchio, RM3); and (2) vector-based PRF, where the query embedding is updated by interpolating it with the embeddings of the top retrieved documents. Both approaches require manually designed mechanisms—either for expansion term selection or for embedding fusion—and do not involve end-to-end optimization of how such feedback is interpreted.

In contrast, $E^2$RANK does not explicitly modify the query representation through heuristic expansion. Instead, it uses the top-N retrieved documents as implicit relevance feedback inputs and "learns" to interpret these documents jointly in a listwise context through supervision, capturing not only query-related relevance cues but also document–document relational signals. $E^2$RANK models the set-level interactions among these documents, which is essential for ranking.

For comparison, we implemented two classical PRF-style baselines using the model without listwise training (corresponding to "w/o Listwise in Stage II" in Table 6), and applied (i) a text-based listwise prompt, and (ii) a vector-based Rocchio-style fusion of query and document embeddings.

As the results shown and discussed in Section 4.6, merely injecting PRF signals is ineffective or even harmful, and only a model like $E^2$RANK that learns supervised, ranking-aware feature transformations from candidate documents can achieve significant performance gains.

Interestingly, the behavior of $E^2$RANK exhibits characteristics similar to PRF: it benefits most from high-quality top-ranked documents and suffers when noisy or irrelevant documents (e.g., randomly sampled or tail documents) are used as feedback, as shown in Figure 4 and Figure 2. This supports the interpretation that $E^2$RANK uses top-ranked documents as relevance cues in a PRF-like manner. However, unlike classical PRF, $E^2$RANK does not rely on manually designed fusion mechanisms, but instead learns how to utilize these signals through end-to-end listwise supervision. This allows the model to determine how much each feedback document should contribute, what type of signal it provides, and how to incorporate it into the ranking-oriented embedding space, rather than merely enriching the query surface form or interpolating embedding vectors.

In summary, while $E^2$RANK is not a traditional PRF system, it operates under a learned PRF-like mechanism, where top-ranked documents provide weak relevance signals, but the ability to interpret, weight, and operationalize those signals is learned through listwise ranking supervision, rather than manually designed. We believe this inspires a new paradigm that bridges PRF intuition with modern embedding-based ranking.

## E    EFFICIENCY

We listed the detailed latency results in Table 12 and Table 13. For $E^2$RANK, we calculate the latency of encoding documents separately from other latencies, because if we use $E^2$RANK as the retrieval model at the same time, the embedding of the document can be reused to avoid duplicate encoding.

Table 12: Reranking latency per query (s) for $E^2$RANK on the Covid Dataset.

| | Encoding Documents | Others | Overall |
|---|---|---|---|
| $E^2$RANK-0.6B | 0.50 | 0.13 | 0.63 |
| $E^2$RANK-4B | 1.74 | 0.43 | 2.17 |
| $E^2$RANK-8B | 2.76 | 0.64 | 3.40 |

Table 13: Reranking latency per query (s) on the Covid Dataset.

| | Overall Latency |
|---|---|
| Qwen3-0.6B (Pointwise) | 0.40 |
| Qwen3-4B (Pointwise) | 1.39 |
| Qwen3-8B (Pointwise) | 2.32 |
| RankQwen3-0.6B | 4.58 |
| RankQwen3-4B | 11.25 |
| RankQwen3-8B | 16.93 |

## F    ANALYSIS

### F.1    INFLUENCE OF THE RERANKING DEPTH

We conducted a comprehensive set of reranking experiments to evaluate the generalizability and robustness of our approach. Specifically, we varied both the reranking depth (Top-10, Top-20, Top-50, Top-100) and evaluated performance using three standard ranking metrics: NDCG@1, NDCG@5, and NDCG@10. The dataset we used is DL20 and BM25 is served as the first-stage retriever. This setup enables us to analyze not only the ability to correctly identify the single most relevant document, but also the overall relevance distribution across the top-ranked results.

Figure 7 presents a comparison between $E^2$RANK and RankQwen3 across all configurations. A clear trend emerges: $E^2$RANK consistently matches or outperforms RankQwen3 on NDCG@5

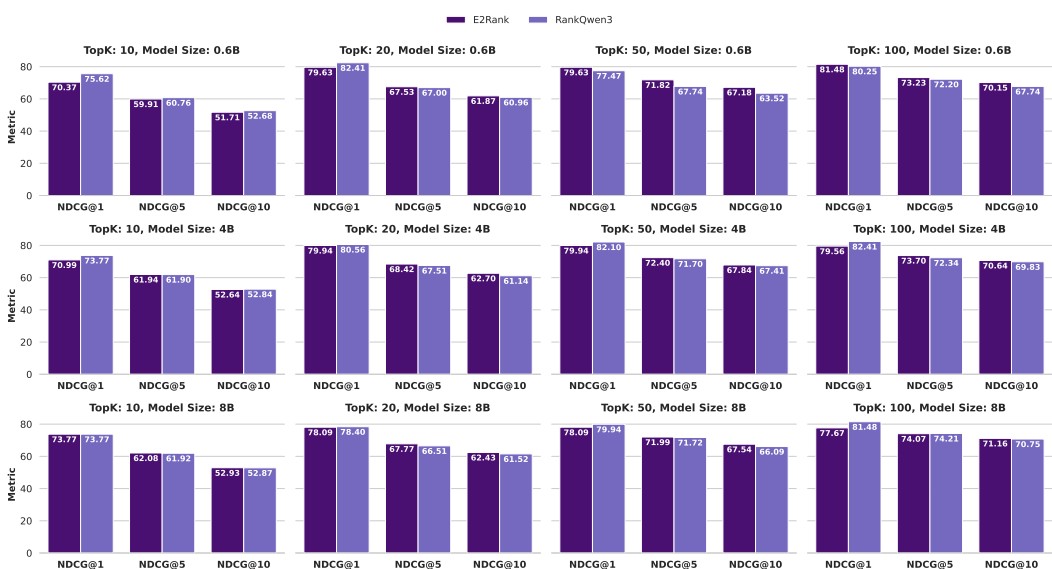

Figure 7: Results of different reranking settings on DL20.

and NDCG@10, highlighting stronger listwise ranking capability and better top-K discrimination. RankQwen3 achieves higher NDCG@1 in several cases—particularly with smaller models or shallower reranking depths. We believe this is consistent with its generative design, which is effective when selecting the most relevant document at the first generative position. In contrast, $E^2$RANK optimizes the distribution of cosine similarity, making it inherently better at capturing global document relevance across the ranked list.

Furthermore, $E^2$RANK exhibits strong scalability, generalizing well across different ranking depths. Importantly, $E^2$RANK maintains competitive precision at rank 1 while delivering stronger performance at broader cutoffs, striking an effective balance between ranking quality and efficiency. These results collectively confirm the robustness, scalability, and practical applicability of $E^2$RANK in real-world reranking scenarios.

### F.2 RERANKING BEHAVIOR

We analyze the behavior of our rerankers along two dimensions: (i) their ability to refine the ranking in the head (top-ranked documents) and (ii) their ability to surface relevant documents from the tail (lower-ranked documents). We use a head cut-off $H$ (typically $H = 20$) to distinguish the "head" (ranked $\leq H$) from the "tail" (ranked $> H$). For the analytical experiments in this section, we reranked the top 100 of BM25 on DL20.

**Promotion and demotion analysis.** To study the interaction between head and tail, we track which documents move into and out of the head when switching from the baseline to a reranker. For each query $q$ and reranker, let $\pi(q, d)$ and $\pi_r(q, d)$ denote the rank positions assigned to $d$ by the first-stage retriever and the reranker, respectively (with $\pi(q, d) = \infty$ if $d$ is not retrieved). We define:

$$P_r(q) = \{d \mid \pi(q, d) > H, \ \pi_r(q, d) \leq H\}$$
$$D_r(q) = \{d \mid \pi(q, d) \leq H, \ \pi_r(q, d) > H\}$$

corresponding to *promoted* and *demoted* documents, respectively. For each reranker, we aggregate basic statistics over all $q$: the total number of promoted or demoted documents, and the proportion of promoted or demoted documents are relevant (with relevance score $> 0$). This reveals whether rerankers tend to replace less relevant head documents with more relevant tail documents.

Table 14 compares how different rerankers modify the head of the ranking by promoting documents from the tail and demoting baseline head documents. Across all models, the fraction of promoted documents with $qrels > 0$ is always higher than for demoted documents . This indicates that all

Table 14: Comparison of document promotion and demotion behavior across rerankers, showing how many documents are promoted from beyond rank 20 into the head, and the relevance quality of promoted versus demoted documents.

| Model | NDCG@10 | #Promoted (Demoted) | % Rel. Promoted | % Rel. Demoted |
|---|---|---|---|---|
| RankQwen3-0.6B | 67.74 | 386 | 0.56 | 0.33 |
| RankQwen3-4B | 69.83 | 405 | 0.61 | 0.31 |
| RankQwen3-8B | 70.75 | 396 | 0.62 | 0.32 |
| $E^2$RANK-0.6B | 70.15 | 515 | 0.57 | 0.26 |
| $E^2$RANK-4B | 70.64 | 527 | 0.56 | 0.27 |
| $E^2$RANK-8B | 71.16 | 518 | 0.58 | 0.26 |

rerankers perform meaningful exchanges, generally replacing less relevant head documents with more relevant tail documents rather than perturbing the ranking arbitrarily.

Within the RankQwen3 family, larger models tend to make slightly higher-quality promotions: the fraction of relevant promoted documents increase from the 0.6B to the 8B, while the relevance of demoted documents decreases slightly.

Compared to RankQwen3, $E^2$RANK promote substantially more documents into the head, indicating a more aggressive reshaping of the top ranks. The relevance profile of these exchanges is still favourable: promoted documents have higher relevance score, while demoted documents are markedly less relevant. In other words, $E^2$RANK performs more head–tail swaps overall, and although each individual promotion is slightly less selective than for the strongest RankQwen3 model, the net effect is to clear out a larger number of low-quality head documents while maintaining a strong bias towards surfacing relevant items from the tail.

**Decomposing head vs. tail contributions to NDCG@100.** To disentangle improvements due to local reranking in the head from those due to promoting tail documents, we decompose the NDCG@100 gain of each reranker relative to the first-stage retriever. We first construct a synthetic *head-only* run that *only* reranks the head, specifically, given the top-100 candidates of each query, rerank the documents ranked $\leq 20$ while keeping the other candidate set fixed. Then, we can define the NDCG@100 gain from only reranking the head, specifically, the difference of NDCG@100 between the head-only run and the original ranking, denoted as $g_{\text{within}}$. Similarly, we can define the NDCG@100 gain of the full run, i.e., the difference of NDCG@100 between the full run and the original ranking, denoted as $g_{\text{total}}$. The NDCG@100 gain from reranking the tail is defined as $g_{\text{tail}} = g_{\text{total}} - g_{\text{within}}$.

Intuitively, $g_{\text{within}}$ measures the gain attributable solely to reranking documents already in the head, while $g_{\text{tail}}$ captures the additional gain from promoting documents from the tail (or, more generally, changing the composition of the head). We report mean and distributional statistics of these gains across queries to characterise each reranker as more "head-refining" or "tail-mining".

Table 15: Decomposition of NDCG@100 gains into contributions from head reordering (within top-20) and tail promotions (beyond rank 20). The NDCG@100 of BM25 is 49.01. The numbers in parentheses represent the percentage improvement in NDCG@100 relative to the BM25 search results.

| Model | NDCG@100 | Mean Gain (Within-20) | Mean Gain (Tail) | Mean Gain (Total) |
|---|---|---|---|---|
| RankQwen3-0.6B | 56.09 | 3.44 (7.0%) | 3.64 (7.4%) | 7.08 (14.4%) |
| RankQwen3-4B | 57.10 | 4.35 (8.9%) | 3.74 (7.6%) | 8.09 (16.5%) |
| RankQwen3-8B | 57.02 | 4.09 (8.3%) | 3.92 (8.0%) | 8.01 (16.3%) |
| E2Rank-0.6B | 57.16 | 4.27 (8.7%) | 3.88 (7.9%) | 8.15 (16.6%) |
| E2Rank-4B | 56.95 | 4.38 (8.9%) | 3.57 (7.3%) | 7.94 (16.2%) |
| E2Rank-8B | 57.51 | 4.56 (9.3%) | 3.94 (8.0%) | 8.50 (17.3%) |

Table 15 reports the decomposition of NDCG@100 improvements into gains from within-top-20 reranking and from promoting relevant documents from the tail. Across all models, the improvement in overall ranking quality is a combination of two complementary effects. The results consistently show that both contributions are substantial and of similar magnitude, confirming that all rerankers are capable of both head refinement and tail mining.

Within each model family, scaling generally improves total NDCG gain. Comparing model families, $E^2$RANK exhibits higher total gains than RankQwen3 at the same scale. Notably, $E^2$RANK tends to achieve slightly higher within-20 gains, indicating stronger head refinement ability, while maintaining competitive tail promotion capability. Meanwhile, RankQwen3 shows a more balanced pattern, with slightly lower head gains but comparable tail gains.

Overall, the results align with our previous observations: RankQwen3 scales toward more selective and precise head–tail exchanges, whereas $E^2$RANK not only makes more extensive replacements in the head but also converts those changes into slightly stronger global effectiveness improvements.

### F.3 TRAINING FROM EXISTING EMBEDDING MODELS

In this work, we mainly chose LLMs (Qwen3 family) primarily because they offer stronger semantic understanding, richer contextual modeling, and longer context length than previous encoder-only models, which we believe is crucial for effectively capturing listwise interactions in the reranking stage. Additionally, it provided checkpoints of different sizes.

However, $E^2$RANK is not restricted to decoder-only models and indeed generally applicable to different embedding backbones. To further validate generality, we have additionally applied our Stage-II training procedure to an existing encoder-based embedding model, GTE-Qwen2-1.5B (based on an LLM but integrated with bidirectional attention mechanisms), without modifying its architecture.

Table 16: Results of training on GTE-Qwen2-1.5B.

| | Q Prompt | DL19 | DL20 | Covid | NFC. | Touche | DBPedia | SciFact | Signal | News | Robust | Avg. | MTEB(v2) |
|---|---|---|---|---|---|---|---|---|---|---|---|---|---|
| | - | 50.58 | 47.96 | 59.47 | 30.75 | 44.22 | 31.80 | 67.89 | 33.05 | 39.52 | 40.70 | 43.43 | - |
| GTE-Qwen2-1.5B | Query-Only | 68.07 | 61.97 | 82.75 | 37.94 | 36.25 | 41.86 | 77.43 | 31.10 | 50.29 | 54.86 | 51.56 | 67.20 |
| + $E^2$RANK Training | Query-Only | 71.31 | 66.20 | 83.52 | 38.48 | 35.38 | 43.56 | 77.16 | 32.68 | 48.65 | 57.63 | 52.13 | 67.19 |
| | Listwise | 69.98 | 68.97 | 81.16 | 39.49 | 46.17 | 41.81 | 75.47 | 34.37 | 53.55 | 56.51 | 53.57 | |

The results in Table 16 show consistent improvements in both reranking benchmarks while maintaining the embedding performance, similar to those observed with the Qwen3 backbone. This supports our claim that the ranking objective and PRF-style listwise prompt are broadly effective and not tied to a particular model class.

## G ADDITIONAL EXPERIMENTAL RESULTS

**Full comparison on BEIR with baselines** We present all detailed results of baselines in Table 17, which is an extended version of Table 2. We report the results of reasoning-intensive rerankers, however, not all of them perform well on these general reranking tasks. In addition, we use the same training dataset with $E^2$RANK to train a cross-encoder style pointwise reranker, using the same RankNet loss. We believe that the reason why them do not perform so well is due to insufficient training data. This comparison between the pointwise model and $E^2$RANK also demonstrates the effectiveness of listwise reranking.

**Full results on MTEB** We present the detailed results on MTEB (eng, v1) benchmark of our models in Table 18. We also evaluate the models on MTEB (eng, v2) benchmark, and the results are shown in Table 19.

**Full results for end-to-end retrieval** We present the detailed results of Table 5 in Table 20 and Table 21.

**Full results for the ablation studies of training** We present the detailed results of Table 6 on Table 22, Table 23, and Table 24.

**Full results for the comparison with PRF baselines** We present the detailed results of Table 7 on Table 25 and Table 26.

Table 17: Full results on BEIR. For reasoning rerankers, the results are borrowed from Liu et al. (2025c) and only contain 7 datasets, excluding Touche2020.

| | Covid | NFCorpus | Touche | DBPedia | SciFact | Signal | News | Robust | Avg. (7) | Avg. (8) |
|---|---|---|---|---|---|---|---|---|---|---|
| BM25 | 59.47 | 30.75 | 44.22 | 31.80 | 67.89 | 33.05 | 39.52 | 40.70 | 43.31 | 43.43 |
| *Previous Fine-tuned Pointwise Reranker* | | | | | | | | | | |
| MonoBERT (340M) | 70.01 | 36.88 | 31.75 | 41.87 | 71.36 | 31.44 | 44.62 | 49.35 | 49.36 | 47.16 |
| MonoT5 (3B) | 79.80 | 37.30 | 32.20 | 48.30 | 58.50 | 76.30 | 32.50 | 44.80 | 53.93 | 51.21 |
| RankT5 (3B) | 81.70 | 37.40 | 31.90 | 49.50 | 58.30 | 77.10 | 38.80 | 45.00 | 55.40 | 52.46 |
| *Previous Fine-tuned Listwise Reranker* | | | | | | | | | | |
| ListT5 (3B) | 84.70 | 37.70 | 33.80 | 53.20 | 57.80 | 77.00 | 33.60 | 46.20 | 55.74 | 53.00 |
| RankVicuna | 79.50 | 32.50 | 33.30 | 45.00 | 47.00 | 68.80 | 32.90 | 44.50 | 50.03 | 47.94 |
| RankZephyr | 83.20 | 37.60 | 32.40 | 44.50 | 74.90 | 31.50 | 52.50 | 54.30 | 54.07 | 51.36 |
| *Zero-shot Listwise Reranker* | | | | | | | | | | |
| RankGPT-4o | 83.41 | 39.67 | 32.26 | 45.56 | 77.41 | 34.20 | 51.92 | 60.25 | 56.06 | 53.09 |
| RankGPT-4o-mini | 80.03 | 38.73 | 30.91 | 44.54 | 73.14 | 33.64 | 50.91 | 57.41 | 54.06 | 51.16 |
| RankQwen3-14B | 84.45 | 38.94 | 38.30 | 44.52 | 78.64 | 33.58 | 51.24 | 59.66 | 55.86 | 53.67 |
| RankQwen3-32B | 83.48 | 39.22 | 37.13 | 45.00 | 78.22 | 32.12 | 51.08 | 60.74 | 55.69 | 53.37 |
| *Reasoning Reranker* | | | | | | | | | | |
| Rank-R1 (7B) | 83.71 | 38.94 | - | 42.27 | 72.16 | 33.08 | 50.60 | 54.46 | 53.60 | - |
| Rank-R1 (14B) | 84.63 | 38.58 | - | 44.05 | 75.96 | 32.95 | 49.20 | 56.91 | 54.61 | - |
| Rank1 (7B) | 79.04 | 37.52 | - | 35.79 | 73.32 | 25.41 | 47.67 | 57.11 | 50.84 | - |
| Rearank (7B) | 81.28 | 35.20 | - | 45.23 | 75.02 | 36.00 | 51.88 | 57.49 | 54.59 | - |
| ReasonRank (7B) | 82.01 | 39.60 | - | 46.03 | 75.55 | 31.36 | 50.50 | 55.40 | 54.35 | - |
| *Fine-tuned Listwise Reranker based on Qwen3* | | | | | | | | | | |
| RankQwen3-0.6B | 78.35 | 36.41 | 37.54 | 39.19 | 71.01 | 30.96 | 44.43 | 46.31 | 49.52 | 48.03 |
| RankQwen3-4B | 83.91 | 39.88 | 32.66 | 43.91 | 76.37 | 32.15 | 50.81 | 59.36 | 55.20 | 52.38 |
| RankQwen3-8B | 85.37 | 40.05 | 31.73 | 45.44 | 78.96 | 32.48 | 52.36 | 60.72 | 56.48 | 53.39 |
| *Pointwise reranker finetund by RankNet loss based on Qwen3* | | | | | | | | | | |
| Qwen3-0.6B (Pointwise) | 84.01 | 33.13 | 36.89 | 33.07 | 70.27 | 27.28 | 37.53 | 45.58 | 47.27 | 45.97 |
| Qwen3-4B (Pointwise) | 80.40 | 31.58 | 29.92 | 40.84 | 72.09 | 25.98 | 47.56 | 56.60 | 50.72 | 48.12 |
| Qwen3-8B (Pointwise) | 81.02 | 28.36 | 34.05 | 40.10 | 70.55 | 26.17 | 43.91 | 52.27 | 48.91 | 47.05 |
| *Ours* | | | | | | | | | | |
| E$^2$RANK-0.6B | 79.17 | 38.60 | 41.91 | 41.96 | 73.43 | 35.26 | 52.75 | 53.67 | 53.55 | 52.09 |
| E$^2$RANK-4B | 83.30 | 39.20 | 43.16 | 42.95 | 77.19 | 34.48 | 52.71 | 60.16 | 55.71 | 54.14 |
| E$^2$RANK-8B | 84.09 | 39.08 | 42.06 | 43.44 | 77.49 | 34.01 | 54.25 | 60.34 | 56.10 | 54.35 |

**Results of using different first-stage retrieval models**  We evaluate the reranking performance of E$^2$RANK on TREC DL19 and DL20 using different first-stage retrieval models, including popular dense embedding models Contriver (Izacard et al., 2021), BGE-base (Xiao et al., 2023), and Qwen3-Embedding-0.6B (Zhang et al., 2025c), as well as an effective neural sparse retrieval model SPLADE++ED (Formal et al., 2022). The full results are shown in Table 27.

We report the reranking results on BRIGHT, using BM25 and the original query for first-stage retrieval, as presented in Table 28. We also report the reranking results on BRIGHT, using BM25 and the GPT4 reasoning query for first-stage retrieval, as presented in Table 29.

Table 18: Detailed Results on MTEB(eng, v1) Benchmark.

| Task | Qwen3-0.6B | | Qwen3-4B | | Qwen3-8B | |
|---|---|---|---|---|---|---|
| | Stage I | Stage II | Stage I | Stage II | Stage I | Stage II |
| AmazonCounterfactualClassification | 79.72 | 82.21 | 83.52 | 82.63 | 81.90 | 81.84 |
| ArXivHierarchicalClusteringP2P | 57.38 | 58.55 | 57.86 | 56.72 | 57.82 | 58.50 |
| ArXivHierarchicalClusteringS2S | 55.44 | 54.00 | 56.09 | 55.37 | 56.99 | 54.26 |
| ArguAna | 52.88 | 50.56 | 52.13 | 51.28 | 55.25 | 54.41 |
| AskUbuntuDupQuestions | 62.21 | 62.92 | 66.45 | 66.92 | 66.31 | 66.80 |
| BIOSSES | 84.68 | 85.22 | 86.13 | 87.93 | 86.46 | 88.40 |
| Banking77Classification | 79.96 | 80.88 | 83.12 | 83.68 | 84.46 | 85.04 |
| BiorxivClusteringP2P.v2 | 38.50 | 39.28 | 40.11 | 40.21 | 39.06 | 39.31 |
| CQADupstackGamingRetrieval | 56.19 | 55.35 | 61.47 | 61.95 | 61.99 | 62.18 |
| CQADupstackUnixRetrieval | 41.53 | 39.31 | 49.86 | 50.24 | 51.07 | 50.51 |
| ClimateFEVERHardNegatives | 26.37 | 30.80 | 37.85 | 27.07 | 39.99 | 31.90 |
| FEVERHardNegatives | 88.02 | 85.68 | 92.26 | 88.85 | 92.86 | 88.91 |
| FiQA2018 | 38.12 | 40.84 | 50.76 | 49.97 | 52.95 | 52.27 |
| HotpotQAHardNegatives | 53.52 | 68.42 | 61.72 | 73.20 | 64.11 | 75.11 |
| ImdbClassification | 76.66 | 82.57 | 86.57 | 89.97 | 86.10 | 89.39 |
| MTOPDomainClassification | 92.60 | 93.62 | 94.09 | 95.71 | 94.11 | 95.70 |
| MassiveIntentClassification | 72.90 | 72.48 | 76.36 | 76.41 | 76.77 | 77.08 |
| MassiveScenarioClassification | 74.58 | 74.71 | 78.96 | 79.54 | 78.04 | 79.24 |
| MedrxivClusteringP2P.v2 | 33.82 | 35.17 | 34.00 | 34.96 | 34.65 | 35.44 |
| MedrxivClusteringS2S.v2 | 32.61 | 31.19 | 32.19 | 32.58 | 32.06 | 33.37 |
| MindSmallReranking | 30.67 | 29.85 | 32.04 | 31.09 | 32.52 | 31.54 |
| SCIDOCS | 16.87 | 17.85 | 20.14 | 20.77 | 20.47 | 22.32 |
| SICK-R | 79.69 | 79.89 | 81.92 | 82.24 | 82.21 | 82.80 |
| STS12 | 76.75 | 74.12 | 77.48 | 76.03 | 78.88 | 77.65 |
| STS13 | 84.07 | 84.19 | 83.47 | 87.07 | 85.00 | 87.48 |
| STS14 | 78.48 | 78.98 | 79.76 | 82.37 | 81.62 | 83.05 |
| STS15 | 85.99 | 86.25 | 87.41 | 88.96 | 88.46 | 89.45 |
| STS17 | 89.92 | 90.09 | 91.63 | 92.59 | 91.58 | 92.09 |
| STS22.v2 | 60.30 | 65.60 | 62.89 | 67.69 | 64.77 | 68.44 |
| STSBenchmark | 84.39 | 84.75 | 86.82 | 88.73 | 87.06 | 88.69 |
| SprintDuplicateQuestions | 91.15 | 93.49 | 90.64 | 95.63 | 92.45 | 95.07 |
| StackExchangeClustering.v2 | 56.38 | 53.12 | 55.68 | 52.04 | 56.96 | 52.71 |
| StackExchangeClusteringP2P.v2 | 38.91 | 38.95 | 40.34 | 41.36 | 40.82 | 41.72 |
| SummEvalSummarization.v2 | 31.55 | 31.66 | 33.53 | 35.08 | 34.62 | 35.07 |
| TRECCOVID | 70.48 | 81.03 | 81.41 | 81.84 | 78.53 | 82.28 |
| Touche2020Retrieval.v3 | 53.79 | 58.46 | 52.39 | 57.51 | 52.37 | 56.61 |
| ToxicConversationsClassification | 64.42 | 64.99 | 69.56 | 69.32 | 68.68 | 69.59 |
| TweetSentimentExtractionClassification | 66.04 | 66.23 | 64.86 | 65.38 | 63.72 | 63.96 |
| TwentyNewsgroupsClustering.v2 | 44.40 | 38.29 | 42.70 | 44.06 | 47.42 | 42.84 |
| TwitterSemEval2015 | 70.68 | 72.13 | 75.93 | 78.47 | 76.49 | 78.35 |
| TwitterURLCorpus | 85.59 | 86.16 | 86.51 | 87.31 | 86.74 | 87.46 |
| **Average** | 62.40 | 63.41 | 65.33 | 66.12 | 65.96 | 66.56 |

Table 19: Detailed Results on MTEB(eng, v2) Benchmark.

| Task | Qwen3-0.6B | | Qwen3-4B | | Qwen3-8B | |
|------|------------|------|----------|------|----------|------|
| | Stage I | Stage II | Stage I | Stage II | Stage I | Stage II |
| AmazonCounterfactualClassification | 79.72 | 82.21 | 83.52 | 82.63 | 81.90 | 81.84 |
| ArXivHierarchicalClusteringP2P | 57.38 | 58.55 | 57.86 | 56.72 | 57.82 | 58.50 |
| ArXivHierarchicalClusteringS2S | 55.44 | 54.00 | 56.09 | 55.37 | 56.99 | 54.26 |
| ArguAna | 52.88 | 50.56 | 52.13 | 51.28 | 55.25 | 54.41 |
| AskUbuntuDupQuestions | 62.21 | 62.92 | 66.45 | 66.92 | 66.31 | 66.80 |
| BIOSSES | 84.68 | 85.22 | 86.13 | 87.93 | 86.46 | 88.40 |
| Banking77Classification | 79.96 | 80.88 | 83.12 | 83.68 | 84.46 | 85.04 |
| BiorxivClusteringP2P.v2 | 38.50 | 39.28 | 40.11 | 40.21 | 39.06 | 39.31 |
| CQADupstackGamingRetrieval | 56.19 | 55.35 | 61.47 | 61.95 | 61.99 | 62.18 |
| CQADupstackUnixRetrieval | 41.53 | 39.31 | 49.86 | 50.24 | 51.07 | 50.51 |
| ClimateFEVERHardNegatives | 26.37 | 30.80 | 37.85 | 27.07 | 39.99 | 31.90 |
| FEVERHardNegatives | 88.02 | 85.68 | 92.26 | 88.85 | 92.86 | 88.91 |
| FiQA2018 | 38.12 | 40.84 | 50.76 | 49.97 | 52.95 | 52.27 |
| HotpotQAHardNegatives | 53.52 | 68.42 | 61.72 | 73.20 | 64.11 | 75.11 |
| ImdbClassification | 76.66 | 82.57 | 86.57 | 89.97 | 86.10 | 89.39 |
| MTOPDomainClassification | 92.60 | 93.62 | 94.09 | 95.71 | 94.11 | 95.70 |
| MassiveIntentClassification | 72.90 | 72.48 | 76.36 | 76.41 | 76.77 | 77.08 |
| MassiveScenarioClassification | 74.58 | 74.71 | 78.96 | 79.54 | 78.04 | 79.24 |
| MedrxivClusteringP2P.v2 | 33.82 | 35.17 | 34.00 | 34.96 | 34.65 | 35.44 |
| MedrxivClusteringS2S.v2 | 32.61 | 31.19 | 32.19 | 32.58 | 32.06 | 33.37 |
| MindSmallReranking | 30.67 | 29.85 | 32.04 | 31.09 | 32.52 | 31.54 |
| SCIDOCS | 16.87 | 17.85 | 20.14 | 20.77 | 20.47 | 22.32 |
| SICK-R | 79.69 | 79.89 | 81.92 | 82.24 | 82.21 | 82.80 |
| STS12 | 76.75 | 74.12 | 77.48 | 76.03 | 78.88 | 77.65 |
| STS13 | 84.07 | 84.19 | 83.47 | 87.07 | 85.00 | 87.48 |
| STS14 | 78.48 | 78.98 | 79.76 | 82.37 | 81.62 | 83.05 |
| STS15 | 85.99 | 86.25 | 87.41 | 88.96 | 88.46 | 89.45 |
| STS17 | 89.92 | 90.09 | 91.63 | 92.59 | 91.58 | 92.09 |
| STS22.v2 | 60.30 | 65.60 | 62.89 | 67.69 | 64.77 | 68.44 |
| STSBenchmark | 84.39 | 84.75 | 86.82 | 88.73 | 87.06 | 88.69 |
| SprintDuplicateQuestions | 91.15 | 93.49 | 90.64 | 95.63 | 92.45 | 95.07 |
| StackExchangeClustering.v2 | 56.38 | 53.12 | 55.68 | 52.04 | 56.96 | 52.71 |
| StackExchangeClusteringP2P.v2 | 38.91 | 38.95 | 40.34 | 41.36 | 40.82 | 41.72 |
| SummEvalSummarization.v2 | 31.55 | 31.66 | 33.53 | 35.08 | 34.62 | 35.07 |
| TRECCOVID | 70.48 | 81.03 | 81.41 | 81.84 | 78.53 | 82.28 |
| Touche2020Retrieval.v3 | 53.79 | 58.46 | 52.39 | 57.51 | 52.37 | 56.61 |
| ToxicConversationsClassification | 64.42 | 64.99 | 69.56 | 69.32 | 68.68 | 69.59 |
| TweetSentimentExtractionClassification | 66.04 | 66.23 | 64.86 | 65.38 | 63.72 | 63.96 |
| TwentyNewsgroupsClustering.v2 | 44.40 | 38.29 | 42.70 | 44.06 | 47.42 | 42.84 |
| TwitterSemEval2015 | 70.68 | 72.13 | 75.93 | 78.47 | 76.49 | 78.35 |
| TwitterURLCorpus | 85.59 | 86.16 | 86.51 | 87.31 | 86.74 | 87.46 |
| **Average** | 62.40 | 63.41 | 65.33 | 66.12 | 65.96 | 66.56 |

Table 20: Full end-to-end ranking performance on BEIR.

| | | Coivd | NFCorpus | Touche | DBPedia | SciFact | Signal | News | Robust | Avg. |
|---|---|-------|----------|--------|---------|---------|--------|------|--------|------|
| $E^2$RANK-0.6b | Retrieval | 81.03 | 33.80 | 29.96 | 41.36 | 71.12 | 27.97 | 42.85 | 52.71 | 47.60 |
| | + Rerank | 83.33 | 37.62 | 30.87 | 43.68 | 72.95 | 27.94 | 50.03 | 58.89 | 50.66 |
| $E^2$RANK-4b | Retrieval | 81.84 | 38.64 | 27.95 | 47.75 | 78.94 | 27.90 | 49.56 | 64.29 | 52.11 |
| | + Rerank | 84.42 | 41.39 | 33.19 | 47.74 | 78.48 | 27.10 | 52.85 | 67.81 | 54.12 |
| $E^2$RANK-8b | Retrieval | 82.29 | 40.08 | 27.95 | 48.75 | 80.91 | 28.13 | 53.46 | 65.55 | 53.39 |
| | + Rerank | 86.61 | 42.33 | 34.86 | 48.20 | 78.99 | 26.31 | 53.75 | 69.58 | 55.08 |

Table 21: Full end-to-end ranking performance on BRIGHT.

| | | StackExchange | | | | | | | Coding | | Theorem-based | | | Avg. |
|---|---|------|-------|--------|------|------|--------|------|-------|------|------|-------|-------|------|
| | | Bio. | Econ. | Earth. | Psy. | Rob. | Stack. | Sus. | Pony. | LC. | AoPS | TheoQ. | ThoT. | |
| $E^2$RANK-0.6b | Retrieval | 19.9 | 29.8 | 17.5 | 20.7 | 17.3 | 15.4 | 12.3 | 4.2 | 38.9 | 9.1 | 13.7 | 21.7 | 18.4 |
| | + Rerank | 27.1 | 37.4 | 23.1 | 31.0 | 22.4 | 19.3 | 20.0 | 3.7 | 38.8 | 8.9 | 17.9 | 21.5 | 22.6 |
| $E^2$RANK-4b | Retrieval | 35.4 | 42.6 | 23.7 | 34.4 | 24.5 | 22.2 | 22.4 | 7.2 | 43.0 | 11.3 | 33.5 | 34.1 | 27.8 |
| | + Rerank | 43.6 | 49.8 | 29.2 | 43.8 | 29.6 | 32.1 | 31.0 | 4.6 | 40.4 | 10.6 | 36.2 | 34.9 | 32.2 |
| $E^2$RANK-8b | Retrieval | 28.6 | 36.6 | 22.3 | 30.9 | 22.2 | 21.7 | 19.8 | 7.3 | 37.9 | 10.3 | 30.2 | 33.3 | 25.1 |
| | + Rerank | 39.9 | 46.6 | 28.9 | 41.7 | 28.3 | 29.7 | 34.4 | 6.1 | 37.4 | 9.1 | 33.6 | 36.7 | 31.0 |

Table 22: Full results of ablation study on BEIR.

| | Coivd | NFCorpus | Touche | DBPedia | SciFact | Signal | News | Robust | Avg. |
|---|---|---|---|---|---|---|---|---|---|
| BM25 | 59.47 | 30.75 | 44.22 | 31.80 | 67.89 | 33.05 | 39.52 | 40.70 | 43.43 |
| E$^2$RANK-0.6B | 79.17 | 38.60 | 41.91 | 41.96 | 73.43 | 35.26 | 52.75 | 53.67 | 52.09 |
| w/o Stage I | 79.22 | 38.13 | 40.98 | 40.99 | 73.18 | 33.35 | 51.74 | 53.01 | 51.33 |
| w/o InfoNCE in Stage II | 79.48 | 39.02 | 40.87 | 42.27 | 74.27 | 34.42 | 52.44 | 54.59 | 52.17 |
| w/ only Stage I | 77.04 | 35.83 | 25.35 | 40.58 | 70.08 | 31.91 | 43.95 | 45.72 | 46.31 |
| w/o RankNet in Stage II | 80.85 | 36.27 | 32.67 | 40.93 | 71.95 | 31.69 | 47.55 | 51.98 | 49.24 |
| w/o Listwise in Stage II | 81.47 | 36.79 | 33.33 | 41.41 | 72.44 | 31.93 | 48.83 | 53.24 | 49.93 |

Table 23: Full results of ablation study on BRIGHT.

| | StackExchange | | | | | | | Coding | | Theorem-based | | | Avg. |
|---|---|---|---|---|---|---|---|---|---|---|---|---|---|
| | Bio. | Econ. | Earth. | Psy. | Rob. | Stack. | Sus. | Pony. | LC. | AoPS | TheoQ. | ThoT. | |
| ReasonIR | 43.5 | 43.0 | 32.8 | 38.9 | 21.1 | 30.6 | 27.3 | 31.6 | 19.6 | 7.3 | 34.1 | 36.7 | 30.5 |
| E$^2$RANK-0.6B | 44.1 | 46.5 | 31.0 | 40.8 | 26.1 | 30.6 | 30.6 | 11.7 | 38.5 | 8.0 | 35.9 | 28.0 | 31.0 |
| w/o Stage I | 44.4 | 46.9 | 29.9 | 40.7 | 25.8 | 26.5 | 32.0 | 13.8 | 37.5 | 8.1 | 31.5 | 30.8 | 30.7 |
| w/o InfoNCE in Stage II | 42.2 | 45.0 | 27.4 | 41.4 | 25.5 | 29.9 | 29.0 | 10.4 | 36.8 | 7.1 | 36.1 | 29.1 | 30.0 |
| w/ only Stage I | 11.1 | 16.7 | 13.0 | 13.3 | 14.8 | 11.0 | 10.3 | 10.0 | 37.4 | 9.0 | 14.4 | 22.7 | 15.3 |
| w/o RankNet in Stage II | 25.1 | 36.2 | 22.0 | 26.2 | 19.8 | 20.1 | 16.2 | 8.0 | 40.2 | 9.7 | 20.2 | 25.0 | 22.4 |
| w/o Listwise in Stage II | 25.1 | 36.5 | 22.1 | 27.0 | 20.1 | 20.4 | 16.9 | 8.4 | 40.1 | 10.1 | 20.7 | 25.0 | 22.7 |

Table 24: Detailed Results of ablation study on MTEB(eng, v2) Benchmark.

| Task | E$^2$RANK-0.6B | w/o Stage I | w/o InfoNCE in Stage II | w/ only Stage I | w/o RankNet in Stage II | w/o Listwise in Stage II |
|---|---|---|---|---|---|---|
| AmazonCounterfactualClassification | 82.21 | 70.34 | 80.76 | 79.72 | 81.6 | 81.18 |
| ArXivHierarchicalClusteringP2P | 58.55 | 57.87 | 57.09 | 57.38 | 58.52 | 58.13 |
| ArXivHierarchicalClusteringS2S | 54.00 | 54.23 | 55.77 | 55.44 | 54.74 | 54.66 |
| ArguAna | 50.56 | 51.46 | 50.85 | 52.88 | 49.04 | 49.59 |
| AskUbuntuDupQuestions | 62.92 | 61.94 | 61.21 | 62.21 | 62.72 | 62.98 |
| BIOSSES | 85.22 | 85.35 | 85.13 | 84.68 | 85.54 | 85.64 |
| Banking77Classification | 80.88 | 79.99 | 81.1 | 79.96 | 80.82 | 81.23 |
| BiorxivClusteringP2P.v2 | 39.28 | 38.63 | 38.27 | 38.50 | 38.82 | 40.17 |
| CQADupstackGamingRetrieval | 55.35 | 53.32 | 55.36 | 56.19 | 56.33 | 57.02 |
| CQADupstackUnixRetrieval | 39.31 | 38.3 | 39.27 | 41.53 | 40.42 | 40.6 |
| ClimateFEVERHardNegatives | 30.80 | 28.91 | 28.03 | 26.37 | 30.53 | 30.91 |
| FEVERHardNegatives | 85.68 | 76.43 | 63.58 | 88.02 | 86.17 | 85.35 |
| FiQA2018 | 40.84 | 36.79 | 34.74 | 38.12 | 40.88 | 40.91 |
| HotpotQAHardNegatives | 68.42 | 63.33 | 59.29 | 53.52 | 67.8 | 69.22 |
| ImdbClassification | 82.57 | 73.02 | 82.01 | 76.66 | 80.73 | 80.91 |
| MTOPDomainClassification | 93.62 | 92.75 | 93.67 | 92.60 | 93.61 | 93.84 |
| MassiveIntentClassification | 72.48 | 69.5 | 71.78 | 72.90 | 72.35 | 72.28 |
| MassiveScenarioClassification | 74.71 | 72.92 | 74.9 | 74.58 | 74.57 | 74.82 |
| MedrxivClusteringP2P.v2 | 35.17 | 35.21 | 34.47 | 33.82 | 34.98 | 36.07 |
| MedrxivClusteringS2S.v2 | 31.19 | 30.82 | 32.17 | 32.61 | 31.18 | 32.04 |
| MindSmallReranking | 29.85 | 29.95 | 30.22 | 30.67 | 30.17 | 30.15 |
| SCIDOCS | 17.85 | 17.07 | 18.13 | 16.87 | 17.63 | 18.09 |
| SICK-R | 79.89 | 70.59 | 80.63 | 79.69 | 79.81 | 79.9 |
| STS12 | 74.12 | 63.39 | 75.15 | 76.75 | 74.28 | 74.28 |
| STS13 | 84.19 | 80.41 | 84.93 | 84.07 | 83.83 | 84.83 |
| STS14 | 78.98 | 74.29 | 79.15 | 78.48 | 78.86 | 79.2 |
| STS15 | 86.25 | 82.54 | 86.57 | 85.99 | 86.41 | 86.68 |
| STS17 | 90.09 | 83.99 | 90.42 | 89.92 | 90.01 | 90.17 |
| STS22.v2 | 65.60 | 65.08 | 65.53 | 60.30 | 62.9 | 63.78 |
| STSBenchmark | 84.75 | 79.05 | 85.46 | 84.39 | 84.58 | 84.88 |
| SprintDuplicateQuestions | 93.49 | 94.73 | 92.67 | 91.15 | 93.78 | 93.82 |
| StackExchangeClustering.v2 | 53.12 | 53.99 | 52.16 | 56.38 | 52.82 | 54.59 |
| StackExchangeClusteringP2P.v2 | 38.95 | 39.1 | 38.94 | 38.91 | 39.19 | 39.77 |
| SummEvalSummarization.v2 | 31.66 | 28.75 | 31.63 | 31.55 | 31.12 | 31.02 |
| TRECCOVID | 81.03 | 78.78 | 67.55 | 70.48 | 81.11 | 82.01 |
| Touche2020Retrieval.v3 | 58.46 | 56.42 | 51.36 | 53.79 | 59.77 | 58.44 |
| ToxicConversationsClassification | 64.99 | 61.35 | 65.41 | 64.42 | 64.52 | 64.91 |
| TweetSentimentExtractionClassification | 66.23 | 62.33 | 66.38 | 66.04 | 66.08 | 65.81 |
| TwentyNewsgroupsClustering.v2 | 38.29 | 41.01 | 41.35 | 44.40 | 39.43 | 42.63 |
| TwitterSemEval2015 | 72.13 | 65.49 | 69.93 | 70.68 | 72.08 | 71.41 |
| TwitterURLCorpus | 86.16 | 85.71 | 85.61 | 85.59 | 86.23 | 86.14 |
| **Average** | 63.41 | 60.61 | 61.92 | 62.40 | 63.31 | 63.66 |

Table 25: Full results of PRF baselines on BEIR.

| | Coivd | NFCorpus | Touche | DBPedia | SciFact | Signal | News | Robust | Avg. |
|---|---|---|---|---|---|---|---|---|---|
| BM25 | 59.47 | 30.75 | 44.22 | 31.80 | 67.89 | 33.05 | 39.52 | 40.70 | 43.43 |
| $E^2$RANK-0.6B | 79.17 | 38.60 | 41.91 | 41.96 | 73.43 | 35.26 | 52.75 | 53.67 | 52.09 |
| w/o Listwise Prompt | 81.35 | 36.27 | 33.50 | 41.09 | 71.88 | 31.70 | 47.15 | 52.70 | 49.46 |
| w/o Listwise in Stage II | 81.47 | 36.79 | 33.33 | 41.41 | 72.44 | 31.93 | 48.83 | 53.24 | 49.93 |
| + text-based PRF | 75.30 | 34.44 | 34.46 | 37.59 | 67.59 | 33.06 | 48.88 | 40.85 | 46.52 |
| + vector-based PRF | 78.73 | 37.08 | 38.06 | 40.66 | 70.50 | 30.61 | 48.12 | 49.84 | 49.20 |

Table 26: Full results of PRF baselines on BRIGHT.

| | StackExchange | | | | | | | Coding | | Theorem-based | | | Avg. |
|---|---|---|---|---|---|---|---|---|---|---|---|---|---|
| | Bio. | Econ. | Earth. | Psy. | Rob. | Stack. | Sus. | Pony. | LC. | AoPS | TheoQ. | ThoT. | |
| ReasonIR | 43.5 | 43.0 | 32.8 | 38.9 | 21.1 | 30.6 | 27.3 | 31.6 | 19.6 | 7.3 | 34.1 | 36.7 | 30.5 |
| $E^2$RANK-0.6B | 44.1 | 46.5 | 31.0 | 40.8 | 26.1 | 30.6 | 30.6 | 11.7 | 38.5 | 8.0 | 35.9 | 28.0 | 31.0 |
| w/o Listwise Prompt | 24.9 | 34.6 | 21.0 | 25.7 | 17.7 | 18.1 | 15.2 | 7.0 | 38.5 | 9.4 | 20.8 | 25.1 | 21.5 |
| w/o Listwise in Stage II | 25.1 | 36.5 | 22.1 | 27.0 | 20.1 | 20.4 | 16.9 | 8.4 | 40.1 | 10.1 | 20.7 | 25.0 | 22.7 |
| + text-based PRF | 49.3 | 49.4 | 31.2 | 22.0 | 22.0 | 30.6 | 27.0 | 26.5 | 33.9 | 7.0 | 34.2 | 22.5 | 29.6 |
| + vector-based PRF | 26.0 | 45.3 | 20.5 | 29.0 | 16.1 | 21.9 | 17.5 | 8.0 | 35.8 | 6.9 | 23.3 | 12.0 | 21.8 |

Table 27: Reranking results using different first-stage retrievers.

| | BGE-base | | Contriver | | SPLADE++ED | | Qwen3E-0.6B | |
|---|---|---|---|---|---|---|---|---|
| | DL19 | DL20 | DL19 | DL20 | DL19 | DL20 | DL19 | DL20 |
| First-stage Retrieval | 70.22 | 66.21 | 62.02 | 63.42 | 73.08 | 71.97 | 68.05 | 66.69 |
| RankQwen3-0.6B | 72.60 | 72.51 | 68.63 | 71.78 | 75.82 | 74.34 | 74.00 | 72.65 |
| $E^2$RANK-0.6B | 74.53 | 73.97 | 71.81 | 74.52 | 76.04 | 77.82 | 74.83 | 73.42 |
| RankQwen3-4B | 72.71 | 76.31 | 70.89 | 76.06 | 75.56 | 74.78 | 72.42 | 73.29 |
| $E^2$RANK-4B | 75.46 | 74.90 | 72.71 | 76.01 | 75.74 | 79.25 | 74.92 | 74.88 |
| RankQwen3-8B | 73.73 | 75.68 | 72.62 | 75.94 | 74.61 | 75.81 | 73.96 | 75.26 |
| $E^2$RANK-8B | 74.15 | 76.40 | 73.77 | 75.04 | 77.37 | 80.08 | 74.97 | 75.24 |

Table 28: Reranking results on BRIGHT. We use BM25 as the first-stage retriever and use original queries to obtain the top-100 candidates. The baseline results are mainly borrowed from Cai et al. (2025). RankQwen3-14B (32B) are zero-shot, others are all fine-tuned.

| | StackExchange | | | | | | | Coding | | Theorem-based | | | Avg. |
|---|---|---|---|---|---|---|---|---|---|---|---|---|---|
| | Bio. | Econ. | Earth. | Psy. | Rob. | Stack. | Sus. | Pony. | LC. | AoPS | TheoQ. | ThoT. | |
| BM25 | 18.2 | 27.9 | 16.4 | 13.4 | 10.9 | 16.3 | 16.1 | 4.3 | 24.7 | 6.5 | 2.1 | 7.3 | 13.7 |
| *Non-reasoning Listwise Reranker* | | | | | | | | | | | | | |
| RankZephyr | 21.9 | 23.7 | 14.4 | 10.3 | 7.6 | 13.7 | 16.6 | 6.5 | 24.7 | 6.8 | 2.0 | 7.3 | 13.0 |
| RankQwen3-0.6B | 21.2 | 32.3 | 17.4 | 20.8 | 14.7 | 14.9 | 18.8 | 6.0 | 26.6 | 6.4 | 4.5 | 8.7 | 16.0 |
| RankQwen3-4B | 28.8 | 37.4 | 19.2 | 31.4 | 20.5 | 21.8 | 26.8 | 10.0 | 22.5 | 6.3 | 11.5 | 10.8 | 20.6 |
| RankQwen3-8B | 29.7 | 40.2 | 21.0 | 31.0 | 23.3 | 23.2 | 27.0 | 10.1 | 16.9 | 6.5 | 12.0 | 11.6 | 21.1 |
| RankQwen3-14B | 30.7 | 41.3 | 23.4 | 30.1 | 24.7 | 21.1 | 27.4 | 7.5 | 30.0 | 8.9 | 12.0 | 11.7 | 22.4 |
| RankQwen3-32B | 31.9 | 45.5 | 23.8 | 33.2 | 25.6 | 22.5 | 30.8 | 7.5 | 29.7 | 10.9 | 11.7 | 13.0 | 23.8 |
| *Reasoning-Intensive Reranker* | | | | | | | | | | | | | |
| Rank-R1-7B | 26.0 | 28.5 | 17.2 | 24.2 | 19.1 | 10.4 | 24.2 | 4.3 | 19.8 | 4.3 | 10.9 | 8.3 | 16.4 |
| Rank1-7B | 31.6 | 34.4 | 18.0 | 23.5 | 16.7 | 18.6 | 22.9 | 20.1 | 9.4 | 4.5 | 9.4 | 9.9 | 18.3 |
| Rearank-7B | 23.4 | 27.4 | 18.5 | 24.2 | 17.4 | 16.3 | 25.1 | 8.0 | 27.0 | 7.4 | 9.5 | 7.9 | 17.7 |
| JudgeRank-8B | 28.7 | 32.2 | 20.9 | 24.6 | 16.5 | 18.3 | 20.6 | 11.7 | 7.1 | 4.7 | 8.4 | 10.0 | 17.0 |
| ERank-4B | 30.4 | 42.5 | 21.5 | 27.7 | 22.4 | 22.9 | 24.0 | 31.6 | 14.6 | 11.0 | 12.1 | 11.4 | 22.7 |
| ERank-14B | 31.2 | 43.6 | 25.8 | 27.8 | 23.1 | 23.9 | 24.6 | 29.8 | 16.8 | 8.6 | 10.5 | 11.9 | 23.1 |
| *Ours* | | | | | | | | | | | | | |
| $E^2$RANK-0.6B | 27.1 | 41.7 | 20.7 | 24.3 | 19.8 | 22.1 | 19.6 | 4.8 | 32.7 | 10.7 | 8.6 | 9.9 | 20.2 |
| $E^2$RANK-4B | 27.8 | 45.6 | 23.9 | 27.6 | 21.4 | 25.0 | 24.3 | 5.0 | 34.8 | 12.3 | 9.3 | 10.8 | 22.3 |
| $E^2$RANK-8B | 28.7 | 45.2 | 24.4 | 27.2 | 22.9 | 25.2 | 25.6 | 6.6 | 32.5 | 11.8 | 9.3 | 10.7 | 22.5 |

Table 29: Reranking results on BRIGHT. We use BM25 as the first-stage retriever and use GPT4 reasoning-queries to obtain the top-100 candidates. The baseline results are mainly borrowed from Cai et al. (2025).

| | StackExchange | | | | | | | Coding | | Theorem-based | | | Avg. |
|---|---|---|---|---|---|---|---|---|---|---|---|---|---|
| | Bio. | Econ. | Earth. | Psy. | Rob. | Stack. | Sus. | Pony. | LC. | AoPS | TheoQ. | ThoT. | |
| BM25 | 18.2 | 27.9 | 16.4 | 13.4 | 10.9 | 16.3 | 16.1 | 4.3 | 24.7 | 6.5 | 2.1 | 7.3 | 13.7 |
| *Non-reasoning Listwise Reranker* | | | | | | | | | | | | | |
| RankQwen3-0.6B | 26.8 | 50.5 | 46.5 | 23.6 | 37.7 | 21.5 | 27.7 | 26.6 | 19.7 | 21.3 | 4.4 | 22.6 | 20.1 |
| RankQwen3-4B | 31.7 | 49.6 | 47.6 | 25.8 | 44.5 | 27.3 | 30.9 | 37.5 | 31.2 | 20.8 | 6.8 | 32.7 | 25.6 |
| RankQwen3-8B | 31.6 | 50.8 | 47.0 | 26.8 | 45.4 | 28.0 | 30.5 | 38.1 | 27.4 | 20.0 | 6.4 | 32.4 | 26.8 |
| *Reasoning-Intensive Reranker* | | | | | | | | | | | | | |
| Rank-R1-7B | 23.9 | 38.2 | 29.4 | 23.4 | 33 | 24.9 | 14.9 | 33.2 | 18.2 | 16.1 | 3.8 | 16.6 | 34.8 |
| Rank1-7B | 25.5 | 45.8 | 37 | 22.2 | 31.7 | 20.6 | 23 | 34.2 | 15.7 | 19.8 | 1.3 | 19.8 | 34.7 |
| Rerank-7B | 29.1 | 42 | 37.5 | 26.4 | 39.1 | 25 | 25.1 | 32.6 | 26.2 | 29.2 | 5.9 | 28 | 32.2 |
| JudgeRank-8B | 24.4 | 41.4 | 34.7 | 26.2 | 36 | 24 | 27.6 | 26.1 | 10.2 | 14.2 | 3.4 | 20.3 | 28.9 |
| ERank-4B | 32.9 | 48.2 | 46.7 | 30 | 43.1 | 28.4 | 31.5 | 38.1 | 28.5 | 23.5 | 10.4 | 26.9 | 39.0 |
| ERank-14B | 33.5 | 51.4 | 48.6 | 30.8 | 41.3 | 26.7 | 35.6 | 39.1 | 27.3 | 26.4 | 10.9 | 25.7 | 37.9 |
| *Ours* | | | | | | | | | | | | | |
| E$^2$RANK-0.6B | 29.6 | 47.6 | 52.2 | 27.4 | 42.0 | 26.8 | 31.5 | 29.5 | 19.5 | 31.8 | 7.5 | 19.2 | 20.5 |
| E$^2$RANK-4B | 32.0 | 49.6 | 51.1 | 31.6 | 43.8 | 29.0 | 33.7 | 33.2 | 16.4 | 31.9 | 7.6 | 33.2 | 22.9 |
| E$^2$RANK-8B | 32.6 | 51.7 | 51.4 | 32.1 | 45.6 | 27.3 | 35.5 | 34.5 | 16.2 | 31.1 | 8.3 | 32.7 | 25.2 |

