# OpenReview forum: "$\text{E}^2\text{Rank}$: Your Text Embedding can Also be an Effective and Efficient Listwise Reranker"
_ICLR.cc/2026/Conference — ICLR 2026 Conference Withdrawn Submission_

### Official Review · Reviewer_wj5W · 2025-10-31

**Soundness:** 2
**Presentation:** 3
**Contribution:** 2
**Rating:** 4
**Confidence:** 4

**Summary:**

This paper proposes a unified framework that enables a single embedding model to handle both retrieval and listwise reranking efficiently. The approach reinterprets listwise prompts as pseudo-relevance feedback (PRF) queries and introduces a two-stage training scheme combining contrastive and listwise ranking objectives, achieving improvements on BEIR and BRIGHT benchmarks with low latency.

**Strengths:**

1. This paper proposes a conceptually idea of using a single embedding model for both retrieval and reranking tasks. Specifically for reranking, the proposed method rely on  a listwise ranking prompt, which is constructed from the original query and its candidate documents, serving as an enhanced query.

2. The experiment demonstrates significantly fewer model size while maintaining competitive accuracy, making it attractive for real-world search applications.

3. This paper reports strong results on BEIR and reasonable performance on BRIGHT, showing that embedding-based reranking can approach LLM listwise ranking effectiveness.

**Weaknesses:**

1. The results of E2RANK-8B on DL19 appear inconsistent — reported as 72.65 in Table 2 but 72.95 in Table 1. Could the authors clarify where this discrepancy arises from?

2. The E2RANK results in Table 5 differ from those presented in earlier tables. Providing a consistent comparison or explicit explanation of these variations would enhance clarity and reproducibility.

3. Even though the proposed model is not trained with listwise prompts in its baseline form, one could still use the same listwise prompt formulation at inference time for reranking. Have the authors evaluated this variant? It would be valuable to understand how much performance improvement arises from the training on listwise prompts versus merely using them at inference, as this could help isolate the contribution of listwise training.

4. About the reference latency, this paper mainly report the model size as the latency. It would be better to include practical runtime cost when compared to the baselines that has the same model size, for example, when comparing E2RANK with RankQ3en3.

5. How does E2RANK scale when applied to substantially larger document collections or real-world web-scale retrieval settings? While the paper emphasizes efficiency, it would be useful to quantify latency under increasing corpus sizes to assess whether the claimed advantages persist at scale.

6. The proposed framework appears adaptable to various embedding backbones (e.g., E5, BGE, or GTR). How sensitive is the overall performance to this choice? Have the authors observed consistent improvements across different base models?

**Questions:**

Please see above weaknesses.

---

> ### Author Response · Authors · 2025-11-22
>
> We would like to thank the reviewer for the valuable comments and insights. We would like to address the points the reviewer raised as follows.
>
> ### **Response to W1**
>
> > The results of E2RANK-8B on DL19 appear inconsistent
>
> Thanks for noticing this discrepancy. This was a typographical error in Table 2. The correct value for E2Rank-8B on DL19 is 72.95, consistent with Table 1. We have corrected the value in the revised manuscript.
>
> ### **Response to W2**
>
> > The E2RANK results in Table 5 differ from those presented in earlier tables.
>
> **The results in Table 5 correspond to a different evaluation setting from the earlier tables, and we have already explained this experimental setup in Section 4.4 in the submitted paper.**
>
> Specifically, Table 5 reports *end-to-end performance*, where E2Rank is used both as the first-stage retriever and as the reranker. In contrast, Tables 1–3 evaluate only the reranking stage while using other retrievers for fairer and broader comparison. Because Table 5 involves a fully unified retrieval–reranking pipeline, the numbers are not expected to match those in the reranking-only setting.
>
> ### **Response to W3**
>
> > Even though the proposed model is not trained with listwise prompts in its baseline form, one could still use the same listwise prompt formulation at inference time for reranking
>
> We appreciate the reviewer’s suggestion. We have indeed evaluated this variant—applying listwise prompts at inference time to the baseline without any listwise training. However, the results were consistently poor. The model struggled to make meaningful use of the listwise context, and its performance was significantly below both E2Rank and even its own query-only baseline.
>
> This is expected: without listwise supervision, the baseline model is not trained to interpret interactions among documents, nor to distinguish relational signals (e.g., comparative relevance) embedded in the listwise prompt. As a result, directly feeding a listwise prompt at inference does not automatically induce listwise ranking behavior.
>
> For comparison, we implemented PRF-style baselines using the same underlying embedding model without listwise training (corresponding to “w/o Listwise in Stage II” in Table 6), and applied a text-based listwise prompt. The results are summarized below (Table 7 in the paper):
>
> |                        | DL20  | BEIR  | BRIGHT |
> |------------------------|-------|-------|--------|
> | E2Rank-0.6B            | 70.15 | 52.09 | 30.96  |
> | E2Rank-0.6B w/o PRF           | 65.25 | 49.46 | 21.50  |
> | Model w/o listwise training  | 66.29 | 49.93 | 22.69  |
> |      +text-based PRF   | 56.57 | 46.52 | 29.62  |
>
> The results demonstrate that **simply injecting PRF signals is insufficient, and the model must be trained to understand how to use these signals**. In contrast, rather than just enriching the query, E2Rank learns how to perform ranking-aware feature transformation conditioned on a set of candidate documents, which requires supervision to learn.

---

> ### Author Response · Authors · 2025-11-22
>
> ### **Response to W4**
>
> > It would be better to include practical runtime cost when compared to the baselines
>
> **We would like to clarify that our latency analysis is not based on model size alone. The paper already reports practical runtime latency, directly comparing E2Rank with RankQwen3 under the same model sizes.** In particular:
>
> - Figure 1(c) presents the measured reranking latency of E2Rank vs. RankQwen3 on the Covid dataset, using the same hardware and the same inference framework.
> - Appendix D (Tables 12 and 13) provides full latency numbers, again showing runtime cost rather than parameter count.
> - Model size is mentioned only to match models of equal scale for a fair comparison; all latency results are based on real execution time.
>
> Because E2Rank avoids autoregressive decoding and reuses precomputed document embeddings, it achieves substantially lower latency than RankQwen3 in these practical measurements, even when both models have identical parameter sizes.
>
> ### **Response to W5**
>
> > How does E2RANK scale when applied to substantially larger document collections or real-world web-scale retrieval settings?
>
> We appreciate the reviewer’s question about scalability to web-scale retrieval. While we are *unable to conduct true web-scale experiments* due to infrastructure constraints, we would like to *clarify how E2Rank behaves theoretically as the corpus size grows*.
>
> **First, the retrieval stage of E2Rank is identical to standard dense retrieval.** Since the model produces a single embedding per query and cosine similarity serves as the scoring function, the computational cost scales in exactly the same way as existing dense retrievers used in large-scale production systems (e.g., BGE, Qwen3-Embedding). Therefore, E2Rank inherits the same sublinear indexing and ANN-search scalability used in real-world deployments.
>
> **Second, the reranking stage of E2Rank operates on only the top-K candidates and will also be efficient.** Because it can reuse the document embedding and only compute the listwise query embedding, its complexity is low and *does not grow with the corpus size*. In other words, E2Rank remains practical as long as the candidate pool is bounded, which is a standard assumption in real systems. Intuitively, it might even be more efficient than the cross-encoder widely used in industry.
>
> ### **Response to W6**
>
> > How sensitive is the overall performance to ...different base models?
>
> Thanks for the insightful question. Our framework is indeed generally applicable to different embedding backbones. In this work, we chose LLMs (Qwen3 family) primarily because they offer stronger semantic understanding, richer contextual modeling, and longer context length than previous encoder-only models, which we believe is crucial for effectively capturing listwise interactions in the reranking stage.
>
> However, E2Rank is not restricted to decoder-only models. To further validate generality, we have additionally applied our Stage-II training procedure to an existing encoder-based embedding model, GTE-Qwen2 (based on an LLM but integrated with bidirectional attention mechanisms), without modifying its architecture.
>
> The results (now included in Appendix F.3) show consistent improvements in both reranking benchmarks while maintaining the embedding performance, similar to those observed with the Qwen3 backbone. **This supports our claim that the ranking objective and PRF-style listwise prompt are broadly effective and not tied to a particular model class.**
>
> |                   | Query Prompt | DL19  | DL20  | BEIR avg | MTEB(eng, v2) |
> |-------------------|--------------|-------|-------|----------|---------------|
> |                   | -            | 50.58 | 47.96 | 43.43    | -             |
> | GTE-Qwen2-1.5B    | Query-Only   | 68.07 | 61.97 | 51.56    | 67.20         |
> | + E2Rank Training | Query-Only   | 71.31 | 66.20 | 52.13    | 67.19         |
> |                   | Listwise     | 69.98 | 68.97 | 53.57    | 67.19         |

---

### Official Review · Reviewer_Gi3W · 2025-10-31

**Soundness:** 2
**Presentation:** 3
**Contribution:** 2
**Rating:** 4
**Confidence:** 5

**Summary:**

This paper proposes E2Rank, a unified framework that enables a single text embedding model to perform both the first-stage retrieval and the second-stage listwise reranking. The core idea is to reinterpret the listwise reranking prompt, which contains the query and top-K candidate documents, as a Pseudo-Relevance Feedback (PRF) query. This "PRF query" is then encoded into an embedding, and reranking is performed efficiently via cosine similarity with pre-computed document embeddings.

**Strengths:**

1. This paper proposes E2Rank, a unified framework that enables a single text embedding model to perform both the first-stage retrieval and the second-stage listwise reranking.
2. The paper provides strong, quantitative evidence of dramatically reduced inference latency compared to auto-regressive listwise baselines.

**Weaknesses:**

1. First of all, the motivation of this paper is somewhat peculiar. The use of a dual-encoder approach for reranking aims to reduce the cost associated with cross-encoder, yet it achieves even better performance than cross-encoder, which is rather peculiar. The effectiveness of pseudo-relevance feedback (PRF) in dense retrieval has already been demonstrated by many retrieval models [1, 2, 3, 4], and it is not the first time it has been used in this paper. A very naive approach to PRF involves concatenating top-retrieval results with the query to enrich the query representation. However, it is somewhat unexpected that using PRF for a dual encoder can directly outperform a cross-encoder (Even with both rank loss and contrastive learning loss).
[1] Yuanhua Lv and ChengXiang Zhai. 2009. A comparative study of methods for estimating query language models with pseudo feedback. In Proceedings of the 18th ACM conference on Information and knowledge management. 1895–1898.
[2] Joseph Rocchio. 1971. Relevance feedback in information retrieval. The Smart retrieval system-experiments in automatic document processing (1971), 313–323.
[3] Guihong Cao, Jian-Yun Nie, Jianfeng Gao, and Stephen Robertson. 2008. Selecting good expansion terms for pseudorelevance feedback. In Proceedings of the 31st annual international ACM SIGIR conference on Research and development in information retrieval. 243–250.
[4] Hang Li, Ahmed Mourad, Shengyao Zhuang, Bevan Koopman, and Guido Zuccon. 2023. Pseudo Relevance Feedback with Deep Language Models and Dense Retrievers: Successes and Pitfalls. ACM Trans. Inf. Syst. 41, 3, Article 62 (July 2023), 40 pages. https://doi.org/10.1145/3570724

2. Secondly, the experimental comparisons do not seem reasonable. For example, in the comparison of MTEB, only last year's models were compared. The paper uses a backbone model from the Qwen3 series. Currently, Qwen3-embedding 8B MTEB eng v2 ranks second on the leaderboard. The Qwen3-embedding model is purely based on a dual-encoder architecture and does not use PRF. The model training utilizes both synthetic data and publicly available data. The Qwen3-embedding technical report also presents results using only publicly available data. However, the paper does not compare with Qwen3-embedding on MTEB. Additionally, the embedding model BGE-en-ICL [1], which shares a similar idea of enriching query expression as this paper, achieves a score of 66.08 on MTEB v1 (56), while the E2Rank in this paper scores 65.03.

Furthermore, in the comparisons on TREC DL and BEIR, the training datasets used for the fine-tuning listwise ranker models (the main baseline: RankQwen, in this work) being compared are relatively weak. For instance, RankQwen utilizes a dataset derived from RankZephyr. The original RankZephyr paper states that it was trained on a labeled dataset of 100,000 queries from MS MARCO. However, why does line 844 in the paper indicate that RankQwen used only 40K queries? In contrast, the E2Rank model employs 1.5M queries in the first stage and 87K queries in the second stage. It is unclear whether the performance improvement of E2Rank over RankQwen in re-ranking tasks stems from the method itself or the training dataset. Simply replacing the backbone without conducting comparisons on training sets of a similar scale raises concerns about the fairness of the experimental comparisons, which need to be addressed.
[1] Li, C., Qin, M., Xiao, S., Chen, J., Luo, K., Shao, Y., ... & Liu, Z. (2024). Making text embedders few-shot learners. arXiv preprint. ICLR26.

3. The paper suffers from minor but noticeable grammatical errors and occasionally awkward phrasing (e.g., "remarkking," "documents as PRF contribute meaningfully").

**Questions:**

Please see the weaknesses.

---

> ### Author Response · Authors · 2025-11-22
>
> We would like to thank the reviewer for the valuable comments and insights. We would like to address the points the reviewer raised as follows.
>
> ### **Response to W1**
>
> > First of all, the motivation of this paper is somewhat peculiar. The use of a dual-encoder approach for reranking aims to reduce the cost associated with cross-encoder, yet it achieves even better performance than cross-encoder, which is rather peculiar.
>
> We understand the reviewer’s concern that it may seem unusual for a dual-encoder style model to outperform a cross-encoder.  However, we would like to clarify that **E2Rank is not a conventional dual encoder, but is an improved listwise reranker.**
>
> **It's important to emphasize that our initial motivation was to improve a listwise reranker**, removing the redundant generation processes while retaining more important interaction information. In fact, it embeds listwise document–document interactions directly into the model through end-to-end training, which enables *richer semantic modeling* than standard pointwise dual-encoders or cross-encoders and aligns more closely with ranking objectives. Therefore, although its score function is the same as dual-encoder, the underlying logic is different: **it indicates that explicitly modeling listwise semantics and document interactions can be more expressive, even without autoregressive decoding**.
>
> > The effectiveness of pseudo-relevance feedback (PRF) in dense retrieval has already been demonstrated...
>
> We agree that PRF has long been studied in both text-based and vector-based settings [1–4]. However, there are differences between these works and E2Rank. Specifically, these methods typically *manually design* how retrieved documents modify the query, treating PRF as a *static query expansion* technique. In contrast, E2Rank does not expand or concatenate documents in a heuristic way. Instead, it *learns how to transform PRF signals into better ranking-aware representations end-to-end*, enabling the model to understand which documents should influence the query representation, how, and to what extent. This is fundamentally different from conventionally engineered PRF, and better suited to reranking where document–document comparative reasoning is crucial.
>
> > However, it is somewhat unexpected that using PRF for a dual encoder can directly outperform a cross-encoder (Even with both rank loss and contrastive learning loss).
>
> For comparison, we implemented two classical PRF-style baselines using the same underlying embedding model without listwise training (corresponding to “w/o Listwise in Stage II” in Table 6), and applied (i) a text-based listwise prompt, and (ii) a vector-based Rocchio-style fusion of query and document embeddings. The results are summarized below (Section 4.6, Table 7 in the paper):
>
> |                        | DL20  | BEIR  | BRIGHT |
> |------------------------|-------|-------|--------|
> | E2Rank-0.6B            | 70.15 | 52.09 | 30.96  |
> | E2Rank-0.6B w/o PRF           | 65.25 | 49.46 | 21.50  |
> | Model w/o listwise training  | 66.29 | 49.93 | 22.69  |
> |      +text-based PRF   | 56.57 | 46.52 | 29.62  |
> |      +vector-based PRF | 63.96 | 49.20 | 21.85  |
>
> While both baselines introduce PRF signals, neither comes close to the performance of E2Rank, and in some cases even harms ranking quality.
>
> This demonstrates that **simply injecting PRF signals is insufficient, and the model must be trained to understand how to use these signals**. In contrast, rather than just enriching the query, E2Rank learns how to perform ranking-aware feature transformation conditioned on a set of candidate documents, and therefore achieves a better ranking effectiveness.

---

> ### Author Response · Authors · 2025-11-22
>
> ### **Response to W2 (1)**
>
> Thank you for the detailed suggestions regarding the baselines.
>
> > the paper does not compare with Qwen3-embedding on MTEB...
>
> We fully agree that Qwen3-embedding is a very strong and closely related baseline, especially given that our backbone also comes from the Qwen3 family. There are two reasons we did not include it in our original experiments. First, the publicly available checkpoint for Qwen3-Embedding uses a large amount of synthetic data, making a direct comparison unfair. Second, the version using only public data did not release a checkpoint, and the paper only reports the results of a 0.6B model on MTEB(eng, v2), while our experiments and the baseline for comparison were mainly conducted on MTEB(eng, v1), making a reasonable comparison with the version trained on publicly available data impossible.
>
> > Additionally, the embedding model BGE-en-ICL [1], which shares a similar idea of enriching query expression as this paper...
>
> We agree that BGE-en-ICL is a highly relevant baseline and acknowledge that BGE-en-ICL improves retrieval embeddings via in-context learning and achieves a strong performance. However, we would also like to point out that a fair comparison should primarily consider the zero-shot BGE-en-ICL setting rather than its few-shot variant. This is because E2Rank does not use any external examples or in-context augmentation at inference time—its embeddings are computed purely from the input text (as noted in line 377 of the paper). We have included the zero-shot results of BGE-en-ICL in the revised manuscript.
>
> **We would also like to emphasize that our goal is not to train yet another state-of-the-art embedding model, and we also did not overclaim this point in the paper**. Instead, the focus of the paper is twofold:
>
> 1. **An embedding model can be further trained to acquire strong listwise reranking ability.** E2Rank achieves competitive reranker-level performance while preserving the efficiency of embedding-based scoring.
> 2. **This training does not harm the model’s embedding quality.** In fact, the retrieval ability often improves, indicating that listwise supervision can be beneficial for both retrieval and reranking within a single unified model.
>
> This aligns with the motivation of the paper, showing that reranking and retrieval can be unified in a single embedding model without sacrificing either efficiency or quality.
>
> ### **Response to W2 (2)**
>
> > the training datasets used for the fine-tuning listwise ranker models being compared are relatively weak...
>
> We appreciate the reviewer’s attention to data control and agree that using comparable training data is important for a fair comparison. In fact, we have also trained E2Rank using the same MS MARCO dataset as RankQwen; this experiment was conducted earlier but was not included in the main submission due to space constraints.h
>
> The mixed dataset reported in the main paper is used to match the practice of modern embedding models, which routinely leverage heterogeneous retrieval data to improve general embedding quality. Since E2Rank is designed as a unified embedding–reranking model, we believe aligning with this practice is appropriate.
>
> **To directly address the question about fairness, we have now added the results where both RankQwen and E2Rank are trained on the same Stage-II dataset**, specifically MS MARCO. These newly included results (summarized in the following table and detailed in Tables 1 and 3 of the paper) show that **E2Rank still outperforms RankQwen overall under fully controlled training data and model size**. Leveraging more extensive datasets further boosts the performance of E2Rank. We also added further clarification and analysis in the revision.
>
> We hope this resolves the concern and demonstrates that the improvements of E2Rank do not merely come from differences in data, but from the proposed unified design and training strategy.
>
> | Model          | Data  | DL19  | DL20  | BEIR Avg. | BRIGHT Avg. |
> |----------------|-------|-------|-------|-----------|-------------|
> |                | -     | 50.58 | 47.96 | 43.43     | 30.5        |
> | RankQwen3-0.6B | MS.   | 69.11 | 67.74 | 48.03     | 29.1        |
> | E2Rank-0.6B    | MS.   | 70.78 | 70.55 | 50.74     | 28.5        |
> | E2Rank-0.6B    | Mixed | 70.84 | 70.15 | 52.09     | 31.0        |
> ||||||||
> | RankQwen3-4B   | MS.   | 72.36 | 69.83 | 52.38     | 32.0        |
> | E2Rank-4B      | MS.   | 70.67 | 71.05 | 52.99     | 31.2        |
> | E2Rank-4B      | Mixed | 70.44 | 70.64 | 54.14     | 32.4        |
> ||||||||
> | RankQwen3-8B   | MS.   | 73.15 | 70.75 | 53.39     | 32.0        |
> | E2Rank-8B      | MS.   | 71.66 | 70.87 | 53.86     | 32.7        |
> | E2Rank-8B      | Mixed | 72.95 | 71.16 | 54.35     | 33.4        |
>
> ### **Response to W3**
>
> We have carefully revised the manuscript to correct all grammatical mistakes, typos, and awkward phrasing to ensure clarity and fluency throughout the paper.

---

### Official Review · Reviewer_xiiK · 2025-11-01

**Soundness:** 3
**Presentation:** 3
**Contribution:** 3
**Rating:** 6
**Confidence:** 5

**Summary:**

This paper introduces a new technique called e2rank where the query where the embedding can be used as a ranker. The key to the technique is that during reranking, the query is augmented with the top-20 documents and re-embed. This gives the query additional context, alleviating typical limitations with pointwise ranking techniques and gives a possible way for embeddings to compete with listwise reranking approaches. There are very strong results, although perhaps lacking on analysis to build intuition why the technique works well. Finally, e2rank seems very dependent on the correct training setup. Fortunately, there are many training details provided in the paper and appendix.

**Strengths:**

S1. The technique is a clever way to integrate more context into the query and improve quality when reranking with embeddings. This could be an appealing efficient alternative to listwise rerankers in some settings.

S2. The results are strong across many datasets, although the paper is lacking in key analysis.

S3. The paper includes results when training, and many details about the training setup.

**Weaknesses:**

W1. The figure 2 shows the technique does not benefit much beyond using the top-20 documents in the listwise prompt and can even get worse after the top-20. This is a similar pattern as shown in previous work on reranker weaknesses where reranker quality starts to degrade as more documents are reranked (Jacob et al, https://arxiv.org/abs/2411.11767), although in e2rank's case it is the size of the context rather than the number of documents. This could indicate e2rank is very sensitive to the training setup, and some other setup could result in better performance, although it could be a natural limitation of the technique.

W2a. More generally, the evaluation is very focused on extracting the top-20 documents when reranking the top-100. It could be that RankQwen3 and other baselines are stronger in other settings, such extracting the top-10 from the top-50, or even the top-1 from the top-10.

W2b. The latency benefits of e2rank are particularly constrained to the extract top-20 from top-100 setting, and will appear less (or more favorable) in other settings. Also, there is no latency comparison to cross-encoders, which would be much faster than RankQwen3. RankQwen3 is a sliding window model, which is very slow compared to other methods.

W3. From a latency perspective, there are key missing baselines. There is no comparison to Parry et al (https://arxiv.org/abs/2405.14589) which is almost a drop-in replacements for RankQwen3, but is much faster and uses a pivot-focused technique similar to e2rank (e2rank essentially uses multiple pivots but Parry et al uses a single pivot). Also, there are a range of other alternative techniques with quality-latency trade-offs such as pairwise rank prompting (PRP), see Oosterhuis et al (https://arxiv.org/abs/2504.12063) for an overview.

W4. The intuition behind e2rank is not sufficiently supported. There are no examples that show how e2rank is acting like PRF --- pseudo-relevance feedback (PRF) would typically refine the initial query using a small number of documents, pushing the query to emphasize topics shown in the top-N documents. If this is how e2rank works, then I would expect less training would be needed achieve similar performance + I would expect some simmilarity to PRF baselines (which none are compared to). Instead, my guess is that e2rank is doing well simply because the query has more context about the top-N retrieved documents can help to disambiguate the results. I am not sure if this really is the case, but there are no results or analysis to justify the intuition as far as I can tell.

W5. Similar to W4, it's hard to distinguish whether the training data or the e2rank algorithm is key to these good results.

**Questions:**

The BRIGHT and ReasonIR papers showed the embeddings are missing fundamental capabilties regarding query length. In the future, would you expect that new embedding models will still need the specialized training here, or it will be an innate property of new embedding models?

Did you consider alternative to choosing top-N for every query? For example, could use more or less depending on the query. Could also do random sampling from the top-100.

Do you notice whether e2rank is better at ranking documents inside the top-20, or beyond the top-20? Not clear whether Figure 3 answers this question.

---

> ### Author Response · Authors · 2025-11-22
>
> We would like to thank the reviewer for the valuable comments and insights. We would like to address the points the reviewer raised as follows.
>
> ### **Response to W1**
>
> We thank the reviewer for this thoughtful observation. While the pattern in Figure 2 superficially resembles the degradation reported in Jacob et al. (2024), the underlying phenomenon is fundamentally different. In Jacob et al., the performance drop occurs when more documents are actually reranked. This issue is typically linked to cross-encoder rerankers and pointwise training, where the distribution of hard negatives becomes problematic as the candidate pool grows.
>
> In our case, however, the number of documents to be reranked is fixed (top-K = 100). Figure 2 only varies the number of documents used to construct the listwise prompt, not the size of the reranking set itself. We believe this is precisely the characteristic that should exist based on the PRF mechanism:
>
> - When the number of PRF documents is small, adding more quickly improves performance since additional relevant signals enrich the query.
> - However, after a certain point, the marginal benefit diminishes, and additional context may introduce noise, causing slight drops. This “contextual saturation” effect has been documented in previous PRF studies and is not an indication of training instability [1].
>
> We agree that sensitivity would be a concern if the model only performed well at the exact number of PRF documents used during training. During Stage-II training, we use 16 PRF documents, but at inference the model achieves its best performance at 20, rather than peaking at 16. This indicates that E2Rank is not overly sensitive to the training context size and generalizes beyond the training configuration.
>
> [1] Hang Li, Ahmed Mourad, Shengyao Zhuang, Bevan Koopman, and Guido Zuccon. 2023. Pseudo Relevance Feedback with Deep Language Models and Dense Retrievers: Successes and Pitfalls. ACM Trans. Inf. Syst. 41, 3, Article 62 (July 2023), 40 pages. https://doi.org/10.1145/3570724
>
> ### **Response to W2 (a)**
>
> We appreciate the reviewer’s suggestion to test more general reranking setups beyond extracting Top-20 from Top-100. We have now evaluated across multiple configurations, varying both (i) the reranking depth (Top-10, Top-20, Top-50, Top-100) and (ii) the evaluation granularity (NDCG@1, @5, @10). The full results are shown in Appendix F.1, and we list part of the results (8B model) in the following table.
>
> | model     | top-K | NDCG@1 | NDCG@5 | NDCG@10 |
> |-----------|-------|--------|--------|---------|
> | RankQwen3 | 100   | 81.48 | 74.21 | 70.75 |
> | E2Rank    |    | 77.67 | 74.07 | 71.16 |
> | RankQwen3 | 50    | 79.94 | 71.72 | 66.09 |
> | E2Rank |     | 78.09 | 71.99 | 67.54 |
> | RankQwen3 | 20    | 78.4  | 66.51 | 61.52 |
> | E2Rank |     | 78.09 | 67.77 | 62.43 |
> | RankQwen3| 10    | 73.77 | 61.92 | 52.87 |
> | E2Rank |    |  73.77 | 62.08 | 52.93 |
>
> Overall, E2Rank consistently matches or outperforms RankQwen3 in NDCG@5 and NDCG@10 across all settings, demonstrating stronger listwise ranking stability and better top-K discrimination. We do observe that RankQwen3 achieves higher NDCG@1 in several settings, which is expected: RankQwen3 is a generation–based reranker that identifies the single best document at the first position. In contrast, E2Rank uses cosine similarity which is naturally not for identifying the top-1 item but for optimizing the full ranking.
>
> These results confirm that E2Rank is not tailored to a specific Top-20-from-Top-100 setting, but generalizes well across different ranking depths and evaluation granularities, maintaining competitive precision at rank 1, while offering stronger performance at broader cutoffs (NDCG@5/@10) and much better efficiency

---

> ### Author Response · Authors · 2025-11-22
>
> ### **Response to W2 (b) and W3**
>
> We thank the reviewer for the insightful suggestions on latency evaluation.
>
> Our work focuses on listwise reranking, and therefore, our primary efficiency comparison is conducted against listwise LLM-based rerankers. Among the publicly available and reproducible listwise rerankers, RankGPT-like models (including RankQwen3) remain the most representative, and they universally adopt sliding-window generation-based decoding, which constitutes the major source of their latency bottleneck. This is why RankQwen3 was used as our main comparison point.
>
> To address the reviewer’s concerns, we have now added cross-encoder latency results (Table 13), which are faster than RankQwen3 and E2Rank. However, if we exclude the time spent encoding the document (i.e., assuming that document embedding is reusable), E2Rank is actually faster than cross-encoders. The key reason is that cross-encoders, despite being faster in per-query inference, still require query–document encoding, leading to computation that scales linearly with K. In contrast, E2Rank performs only one listwise forward encoding and then ranks via cosine similarity. If we consider E2Rank as an end-to-end model that can be used for both retrieval and reranking, as we have emphasized in the paper, this advantage will become even more apparent.
>
> |             | Encoding Documents | Others | Overall |
> |-------------|--------------------|--------|---------|
> | E2Rank-0.6B | 0.50               | 0.13   | 0.63    |
> | E2Rank-4B   | 1.74               | 0.43   | 2.17    |
> | E2Rank-8B   | 2.76               | 0.64   | 3.40    |
>
> |                        | Overall Latency |
> |------------------------|-----------------|
> | Qwen3-0.6B (Pointwise Trained) | 0.40            |
> | Qwen3-4B (Pointwise Trained)   | 1.39            |
> | Qwen3-8B (Pointwise Trained)   | 2.32            |
> | RankQwen3-0.6B         | 4.58            |
> | RankQwen3-4B           | 11.25           |
> | RankQwen3-8B           | 16.93           |
>
> Regarding Parry et al, while these methods improve efficiency over sliding-window generation, they still fundamentally rely on generative decoding. Parry et al. introduce pivot-based conditioning, but still perform token generation over long prompts. In contrast, E2Rank improves efficiency by removing decoding altogether, moving ranking entirely into embedding space. This represents a qualitatively different form of efficiency improvement—architectural rather than prompt-level—and enables unified retrieval and reranking. As for the improved PRP-style methods, since they are restricted to pairwise prompting, we believe these baselines are beyond the scope of this paper.
>
> Finally, on the concern that efficiency gains may only apply to the Top-20-from-Top-100 setting, actually E2Rank scales sublinearly due to embedding reuse and no decoding, making its efficiency more favorable, not less, as K increases.

---

> ### Author Response · Authors · 2025-11-22
>
> ### **Response to W4 & W5**
>
> We partially agree with the reviewer that E2Rank does not perform PRF in the classical sense of explicitly expanding the query using top-retrieved documents with manually designed heuristics. Instead, E2Rank performs **learned listwise feedback**: the model is not provided with an explicit expanded query, but learns to interpret the top-N retrieved candidates as weak feedback signals and to transform them into ranking-aware embeddings. This is fundamentally different from simply concatenating additional context.
>
> For comparison, we implemented two classical PRF-style baselines using the same underlying embedding model without listwise training (corresponding to “w/o Listwise in Stage II” in Table 6), and applied (i) a text-based listwise prompt, and (ii) a vector-based Rocchio-style fusion of query and document embeddings. The results are summarized below (Section 4.6, Table 7 in the paper):
>
> |                        | DL20  | BEIR  | BRIGHT |
> |------------------------|-------|-------|--------|
> | E2Rank-0.6B            | 70.15 | 52.09 | 30.96  |
> | E2Rank-0.6B w/o PRF           | 65.25 | 49.46 | 21.50  |
> | Model w/o listwise training  | 66.29 | 49.93 | 22.69  |
> |      +text-based PRF   | 56.57 | 46.52 | 29.62  |
> |      +vector-based PRF | 63.96 | 49.20 | 21.85  |
>
> While both PRF baselines introduce PRF-like signals, neither comes close to the performance of E2Rank, and in some cases even harms ranking quality.
>
> This demonstrates that **simply injecting PRF signals is insufficient**, and the model must be trained to understand how to use these signals. In contrast, rather than just enriching the query, E2Rank learns how to perform ranking-aware feature transformation conditioned on a set of candidate documents, which requires supervision to learn.
>
> Interestingly, the behavior of \model exhibits characteristics similar to PRF: it benefits most from high-quality top-ranked documents and suffers when noisy or irrelevant documents (e.g., randomly sampled or tail documents) are used as feedback (Regarding response to Q2)
>
> We therefore do not claim that E2Rank is equivalent to classical PRF, nor that it is simply exploiting extra context. Rather, the core contribution is that **listwise prompt can be interpreted as a PRF-like mechanism that can be learned end-to-end and embedded directly into a dual-encoder–style model, enabling a unified and efficient retrieval–reranking pipeline**.

---

> ### Author Response · Authors · 2025-11-22
>
> ### **Response to Q1**
>
> This is an interesting point. We believe that training data remains important and requires additional training for a targted capability. A richer distribution of query lengths might improve the model's generalization ability to handle query lengths. However, this question touches on long-term model evolution and is outside the direct scope of our paper.
>
> ### **Response to Q2**
>
> Thank you for the suggestion. We conducted an ablation where we kept the number of documents fixed but varied which documents were included in the listwise prompt, including "Top-20", "Random 20", "Last 20", and "Query-only". The results are shown in Section 4.6, Figure 4 in the revised paper, and also in the following table.
>
> | Model        | Query-Only | Top-20 | Random-20 | Last-20 |
> |--------------|------------|--------|-----------|---------|
> | E2Rank-0.6B  | 65.25      | 70.15  | 60.87     | 54.72   |
> | E2Rank-4B    | 68.56      | 70.64  | 62.33     | 54.16   |
> | E2Rank-8B    | 69.47      | 71.16  | 62.08     | 56.27   |
>
> Across all model sizes, using **Top-20 clearly yields the best performance**, while using Random-20 or Last-20 significantly harms ranking quality—even worse than using no document context at all in some cases. This suggests that the model relies specifically on relevant top-ranked documents as implicit feedback signals, **consistent with PRF-like behavior**. Low-quality or off-topic documents introduce contradictory or noisy signals, degrading performance. This also explains why E2Rank’s behavior differs from RankQwen3: since E2Rank does not directly compare every candidate document with the query (as cross-encoders do), it relies more on informative head documents to guide embedding-based listwise reasoning.
>
> Regarding dynamically adjusting N per query, we agree that this is promising. One could estimate how many documents to include based on the similarity distribution between the query embedding and retrieved document embeddings (e.g., a confidence threshold). This will be an interesting extension for future work.
>
> ### **Response to Q3**
>
> Thank you for the interesting question. To address this, we have added a more detailed analysis.
>
> Concretely, we define the “head” as documents ranked ≤20 and the “tail” as those ranked >20, and analyze (i) which tail documents are promoted into the head and which head documents are demoted, and (ii) how much of the NDCG@100 gain comes from reordering within the top-20 vs. promoting tail documents. We recommend that reviewers read Appendix F.2 of the paper to obtain complete details and experimental results. The results show that:
>
> - For both RankQwen3 and E2Rank, promoted documents are more relevant than demoted ones, indicating meaningful head–tail exchanges rather than random perturbations.
> - E2Rank promotes more tail documents into the head than RankQwen3, while still maintaining a favorable relevance profile.
> - When decomposing NDCG@100 gains, compared with RankQwen3, E2Rank exhibits slightly higher gains within the top-20 and comparable gains from the tail, leading to a larger total improvement at the same scale.
>
> | Model          | NDCG@100 | Mean Gain (Within-20) | Mean Gain (Tail) | Mean Gain (Total) |
> |----------------|:--------:|:---------------------:|:----------------:|:-----------------:|
> | RankQwen3-0.6B |   56.09  |      3.44 (7.0\%)     |   3.64 (7.4\%)   |   7.08 (14.4\%)   |
> | RankQwen3-4B   |   57.10  |      4.35 (8.9\%)     |   3.74 (7.6\%)   |   8.09 (16.5\%)   |
> | RankQwen3-8B   |   57.02  |      4.09 (8.3\%)     |   3.92 (8.0\%)   |   8.01 (16.3\%)   |
> | E2Rank-0.6B    |   57.16  |      4.27 (8.7\%)     |   3.88 (7.9\%)   |   8.15 (16.6\%)   |
> | E2Rank-4B      |   56.95  |      4.38 (8.9\%)     |   3.57 (7.3\%)   |   7.94 (16.2\%)   |
> | E2Rank-8B      |   57.51  |      4.56 (9.3\%)     |   3.94 (8.0\%)   |   8.50 (17.3\%)   |
>
> In summary, E2Rank is effective both at refining the ranking inside the top-20 and at surfacing relevant documents from beyond the top-20, with a mild bias toward stronger head refinement. We have clarified this in the revised paper and refer readers to the appendix for full statistics.

---

### Official Review · Reviewer_jze9 · 2025-11-03

**Soundness:** 2
**Presentation:** 3
**Contribution:** 2
**Rating:** 2
**Confidence:** 2

**Summary:**

The paper proposes a framework that enables a language model to perform both retrieval and efficient listwise reranking, addressing the high latency of reranking stage in multi-stage search systems. The core contribution is to augment the query with a few retrieved documents and use it to produce an enhance query embedding. Reranking is then performed through efficient cosine similarity between this enhanced query embedding and the document embeddings. The authors conduct experiments on multiple IR benchmarks such as TREC DL, BRIGHT, and MTEB and show competitive performance against existing rerankers while being efficient.

**Strengths:**

- The paper is easy to follow and mostly well written
- The results are reported for multiple standard IR benchmarks
- The paper addresses an important problem of efficiency in the reranking stage

**Weaknesses:**

### W1 - weak experimental setup
- (a) baselines are not training data controlled - RankQwen (the controlled baseline) is trained on a different and smaller dataset from 2 years ago while the proposed method is trained on a more extensive reranking data. The authors could've trained RankQwen on their stage 2 dataset as well?
- (b) The results of RankZephyr and other LLM based rerankers are reported in a sub-optimal setting, RankZephyr using splade as the retrieval model achieves ~80% on DL20 but the results in the paper are reported for BM25 retrieval where it achieves ~70% - this raises the question why did the authors choose the BM25 setting over the more optimal setting with splade retrieval
- (c) similarly, BRIGHT results are reported in the weaker setting without query expansion, BRIGHT paper itself shows much stronger results with reasoned query expansion and ReasonIR's primary results are reported with expanded queries so the choice of not evaluating with query expansion needs justification

### W2 - lack of novelty
The core of the paper is to learn better embeddings using a retrieval augmented query expansion, similar ideas have been proposed [1, 2] - the method adds an additional ranking objective on the embedding learning but that in itself is not a substantial update. Without a clear empirical justification it is hard to justify the contributions of the paper.

### References
[1] RARe: Retrieval Augmented Retrieval with In-context Examples, Tejaswi et al

[2] Contextual Document Embeddings, Morris et al

**Questions:**

1. In stage 2 (which as per my understanding is training on listwise ranking data), how exactly is the infonce loss implemented?

Please see rest of questions in weakness section.

---

> ### Author Response · Authors · 2025-11-22
>
> We would like to thank the reviewer for the valuable comments and insights. We would like to address the points reveiwer raised as follows.
>
> ### **Response to W1 (a)**
>
> > baselines are not training data controlled
>
> We appreciate the reviewer’s attention to data control and agree that using comparable training data is important for a fair comparison. In fact, we have also trained E2Rank using the same MS MARCO dataset as RankQwen; this experiment was conducted earlier but was not included in the main submission due to space constraints.h
>
> The mixed dataset reported in the main paper is used to match the practice of modern embedding models, which routinely leverage heterogeneous retrieval data to improve general embedding quality. Since E2Rank is designed as a unified embedding–reranking model, we believe aligning with this practice is appropriate.
>
> **To directly address the question about fairness, we have now added the results where both RankQwen and E2Rank are trained on the same Stage-II dataset**, specifically MS MARCO. These newly included results (summarized in the following table and detailed in Tables 1 and 3 of the paper) show that **E2Rank still outperforms RankQwen overall under fully controlled training data and model size**. Leveraging more extensive datasets further boosts the performance of E2Rank. We also added further clarification and analysis in the revision.
>
> We hope this resolves the concern and demonstrates that the improvements of E2Rank do not merely come from differences in data, but from the proposed unified design and training strategy.
>
> | Model          | Data  | DL19  | DL20  | BEIR Avg. | BRIGHT Avg. |
> |----------------|-------|-------|-------|-----------|-------------|
> | RankQwen3-0.6B | MS.   | 69.11 | 67.74 | 48.03     | 29.1        |
> | E2Rank-0.6B    | MS.   | 70.78 | 70.55 | 50.74     | 28.5        |
> | E2Rank-0.6B    | Mixed | 70.84 | 70.15 | 52.09     | 31.0        |
> |||||||
> | RankQwen3-4B   | MS.   | 72.36 | 69.83 | 52.38     | 32.0        |
> | E2Rank-4B      | MS.   | 70.67 | 71.05 | 52.99     | 31.2        |
> | E2Rank-4B      | Mixed | 70.44 | 70.64 | 54.14     | 32.4        |
> |||||||
> | RankQwen3-8B   | MS.   | 73.15 | 70.75 | 53.39     | 32.0        |
> | E2Rank-8B      | MS.   | 71.66 | 70.87 | 53.86     | 32.7        |
> | E2Rank-8B      | Mixed | 72.95 | 71.16 | 54.35     | 33.4        |
>
> ### **Response to W1 (b)**
>
> > why did the authors choose the BM25 setting over the more optimal setting with splade retrieval
>
> Thanks for raising this concern. Our choice of BM25 as the first-stage retriever follows the common evaluation protocol **used in most prior LLM-based reranking works**. In this line of research, it is common to align the retrieval model across methods, and BM25 is by far the most widely adopted choice for TREC DL and BEIR. Using BM25 allows fair comparison against a broader set of baselines, including RankGPT, ListT5, monoT5, and RankZephyr, all of which report their primary results under BM25 retrieval.
> While RankZephyr indeed reports a stronger number when paired with SPLADE (~80% on DL20), this corresponds to a different and stronger retrieval setting.
>
> Actually, we have also included SPLADE-based retrieval results for TREC DL in the appendix (Table 27), enabling comparison under the stronger retrieval configuration as well. For reference, we also added the table below. This result indicates that when using Splade as a retrieval model, E2Rank-8B is slightly lower than RankZephyr in performance, but still maintains competitiveness. We believe that this set of experiments will not affect the main conclusion of the paper.
>
> We believe the BM25 setting remains the most appropriate choice for the main comparison, as it ensures methodological consistency with the broader literature, while the SPLADE-based results in the appendix provide the additional comparison you suggested.
>
> ### **Response to W1 (c)**
>
> > BRIGHT results are reported in the weaker setting without query expansion
>
> **We would like to clarify that our main experiments on BRIGHT have already used reasoned query expansion, and we have presented the experiment details in Section 4.2.** Specifically, when employing ReasonIR as the first-stage retriever, we have used the GPT-4 reason-query expansion, as stated in the Dataset paragraph in Section 4.2. Thus, our main BRIGHT results are aligned with the recommended and widely adopted evaluation protocol in prior work.
>
> Regarding the BM25-based experiments, these results appear only in the appendix as auxiliary analysis, since BM25 retrieval quality is indeed relatively poor. For completeness, we additionally ran BM25 with GPT-4 reason-query expansion and also included the results in Appendix G. These results show consistent trends and do not affect the conclusions of the paper.
>
> We hope this clarifies that our evaluation is conducted under the proper expanded-query setting, and we have supplemented the BM25 results for reference.

---

> ### Author Response · Authors · 2025-11-22
>
> ### **Response to W2**
>
> > ...Without a clear empirical justification it is hard to justify the contributions of the paper.
>
> **We believe the comparison with RARe [1] and Contextual Document Embeddings [2] reflects a misunderstanding of our core contribution.** While we acknowledge that they explore the idea of enriching embeddings with additional contextual signals, their mechanisms and objectives differ fundamentally from E2Rank.
>
> **Different mechanisms for enhancing queries:** First, both RARe and CDE are *isolated* embedding-enhancement methods. RARe augments the query embedding using in-context examples drawn from training q–d pairs, and CDE strengthens document embeddings by leveraging other documents within the same batch. These approaches improve the embedding representations without interacting with the actual retrieval pipeline; the augmentation is *static and independent* of the candidate set.
>
> In contrast, E2Rank exploits the two-stage retrieval pipeline itself. The listwise prompt is constructed from the actual top-K candidates retrieved in Stage I, which naturally encodes rich query–document and document–document interactions. This is fundamentally different from using fixed training examples or batch co-occurrence. Our method is therefore *jointly conditioned on the real candidate set*, not on synthetic or global context, which allows the PRF-style signals to be directly optimized toward ranking.
>
> **Different tasks and goals:** Second, the objectives of the methods differ substantially. RARe and CDE both aim to *improve embedding-based retrieval only*. E2Rank is designed to **improve the listwise reranking and unify retrieval and listwise reranking in a single embedding model by offering a unified scoring mechanism**. The additional RankNet loss enables the model to learn ranking-specific discrimination patterns that standard contrastive learning does not capture. Our experiments (Table 6) show that removing the ranking objective leads to a large drop in reranking performance, demonstrating that the ranking supervision is empirically crucial.
>
> For these reasons, E2Rank cannot be considered a minor variant of [1] or [2]. **It introduces a unified retrieval–reranking formulation and demonstrates a substantial empirical improvement.**
>
> ### **Response to Q1**
>
> Basically, we just implemented the basic InfoNCE loss. Although we labeled all documents with a listwise ranking, we still retained the original positive or negative labels. Therefore, we only need to ignore the listwise ranking and use the original labels for computing loss.

---

### Author Response · Authors · 2025-12-01
**Rebuttal Summary and Note to the Incoming Area Chair**

To the newly assigned Area Chair,

We acknowledge the recent data leak announcement and share the community’s regret.

Since reviewers can no longer join discussions or update scores, we sincerely thank them for their initial evaluations, which greatly helped improve our manuscript. We are encouraged that reviews approve the following strengths of our work:

- **Novel approach:** The paper proposed a novel approach to use a single embedding model for both retrieval and listwise reranking (Reviewer xiiK, Gi3W, wj5W).
- **Strong experiment results:** The paper conducted experiments on multiple standard IR benchmarks (Reviewer jze9, xiiK, wj5W), and the proposed E2Rank demonstrates strong effectiveness and high efficiency (Reviewer jze9, Gi3W, wj5W).
- **Good writing:** The paper is well-written, thus it is comprehensive and easy to follow (Reviewer jze9).

Since none of the reviewers have responded, we are especially grateful to the newly assigned Area Chair for taking on this responsibility without the benefit of reviewer dialogue, and for your commitment to maintaining scientific integrity.

To support your evaluation, we have included a “Rebuttal Summary” below, outlining the main revisions and responses addressing the reviewers’ concerns.

**(1) More reasonable baseline comparison:**

We added a direct comparison between the RankQwen and E2Rank trained with the same data, demonstrating E2Rank’s superiority over traditional listwise methods under identical data and model size conditions, thereby improving the fairness and robustness of the experimental evaluation (Reviewer jze9 "W1 (a)" & Gi3W "W2"; Section 4.2, Table 1 & Table 3).

**(2) More detailed analysis experiments:**

We conducted additional experiments in response to the reviewers’ comments and provided a detailed analysis of E2Rank’s PRF-like behavior, including

- comparisons with previous PRF methods (Reviewer xiiK "W4" & Gi3W "W1"; Section 4.6, Table 7),
- the influence of document selection (Reviewer xiiK "Q2"; Section 4.6, Figure 4),
- the performance under different reranking settings (Reviewer xiiK "W2 (a)"; Appendix F.1, Figure 7).

We also expanded Appendix D with an in-depth discussion of the PRF mechanism, clarifying E2Rank’s PRF-like behavior and the necessity of maintaining a unified training–inference pipeline.

**(3) Discussion on efficiency:**

We added further discussion on efficiency, introduced additional baselines (Reviewer xiiK "W2 (b) & W3"), clarified several aspects of the experimental results (Reviewer wj5W "W4"), and provided a theoretical analysis of E2Rank’s scalability to large-scale applications (Reviewer wj5W "W5").

**(4) Clarifying Misunderstandings:**

We clarified several misunderstandings raised by the reviewers, particularly regarding the experimental setup (Reviewer jze9 "W1 (b)(c)" & wj5W "W2") and the core motivation and contributions of the paper (Reviewer jze9 "W2" & Gi3W "W1"). We believe that resolving these misunderstandings will greatly improve reviewers' acceptance of our paper.

We believe these revisions and responses, particularly the inclusion of a more reasonable baseline and more detailed analyses, solidly validate E2Rank’s efficacy and novelty.

Best regards,

The Authors

---

### Note · Authors · 2026-01-23

**Comment:**

We would like to withdraw this submission from ICLR 2026 and apologize for any inconvenience caused. We sincerely thank the reviewers and the Area Chair for their time and effort.

**Withdrawal Confirmation:**

I have read and agree with the venue's withdrawal policy on behalf of myself and my co-authors.